# Preclinical evaluation of the SARS-CoV-2 M^pro inhibitor RAY1216 shows improved pharmacokinetics compared with nirmatrelvir

Xiaoxin Chen[1,2,13], Xiaodong Huang [3,4,13], Qinhai Ma[3,13], Petr Kuzmič [5,13], Biao Zhou[4,6], Sai Zhang[6], Jizheng Chen[6], Jinxin Xu [4], Bin Liu[3], Haiming Jiang[3], Wenjie Zhang[3], Chunguang Yang[3], Shiguan Wu[3], Jianzhou Huang[2], Haijun Li[2], Chaofeng Long[2], Xin Zhao[7], Hongrui Xu[4], Yanan Sheng[4,8], Yaoting Guo[4], Chuanying Niu[4,8], Lu Xue[4], Yong Xu[4], Jinsong Liu [4], Tianyu Zhang [4], James Spencer [9], Zhenzhen Zhu[10], Wenbin Deng [1], Xinwen Chen[6,11], Shu-Hui Chen [10] ✉, Nanshan Zhong [3,6,12] ✉, Xiaoli Xiong [4] ✉ & Zifeng Yang [3,6,12] ✉

Although vaccines are available for SARS-CoV-2, antiviral drugs such as nirmatrelvir are still needed, particularly for individuals in whom vaccines are less effective, such as the immunocompromised, to prevent severe COVID-19. Here we report an α-ketoamide-based peptidomimetic inhibitor of the SARS-CoV-2 main protease (M^pro), designated RAY1216. Enzyme inhibition kinetic analysis shows that RAY1216 has an inhibition constant of 8.4 nM and suggests that it dissociates about 12 times slower from M^pro compared with nirmatrelvir. The crystal structure of the SARS-CoV-2 M^pro:RAY1216 complex shows that RAY1216 covalently binds to the catalytic Cys145 through the α-ketoamide group. In vitro and using human ACE2 transgenic mouse models, RAY1216 shows antiviral activities against SARS-CoV-2 variants comparable to those of nirmatrelvir. It also shows improved pharmacokinetics in mice and rats, suggesting that RAY1216 could be used without ritonavir, which is co-administered with nirmatrelvir. RAY1216 has been approved as a single-component drug named 'leritrelvir' for COVID-19 treatment in China.

SARS-CoV-2 has become established in the human population through the coronavirus disease 2019 (COVID-19) pandemic and is likely to remain in circulation. Owing to multinational efforts, vaccines were rapidly rolled out in the early stage of the pandemic and proved successful in saving lives. However, probably due to population immune pressures established by infections and vaccinations, SARS-CoV-2 Omicron variants with highly mutated spike proteins quickly emerged[1]. Rapid emergence of highly mutated variants has shown the extraordinary capacity of the virus to escape humoral immunity, representing a great challenge to vaccines and therapeutic antibodies[2,3].

A number of small-molecule SARS-CoV-2 therapeutics have been developed[4]. This therapeutic strategy may be part of a solution to combat SARS-CoV-2 immune escape. Of note, the orally available drugs molnupiravir and Paxlovid have been approved for COVID-19 treatment

**Fig. 1 | Chemical structures of RAY1216 and related antiviral protease inhibitors. a**, The chemical structure of RAY1216; P1′ denotes the warhead moiety and P1–P4 denote the other chemical moieties. **b**, The chemical structure of SARS-CoV-2 M^pro inhibitor PF-07321332 (nirmatrelvir). **c,d**, Chemical structures of telaprevir (**c**) and boceprevir (**d**), both of which inhibit HCV NS3/4A protease.

after being validated through clinical trials. Molnupiravir (LAGEVRIO, also known as EIDD-2801) is a prodrug of *N*-hydroxycytidine; this mutagenic ribonucleoside is a broad-spectrum antiviral agent targeting the viral RNA polymerase by lethal mutagenesis. However, this molecule has also been shown to be mutagenic to the host[5]. Paxlovid is a combination of PF-07321332 (nirmatrelvir) and ritonavir. PF-07321332 is a peptidomimetic that selectively inhibits the SARS-CoV-2 main protease (M^pro, also known as 3C-like protease (3CL^pro))[6,7], while ritonavir is a cytochrome P450 inhibitor that functions to slow down cytochrome-P450-mediated metabolism of PF-07321332 to improve bioavailability. However, the usage of ritonavir limits the clinical application range of Paxlovid owing to the drug–drug interaction, which may cause potential safety issues. Therefore, our original goal is to aim for a drug candidate endowed with a longer half-life while maintaining good enzyme inhibitory potency as shown by PF-07321332. We expect that such a newly designed M^pro inhibitor may possess prolonged pharmacokinetic stability in humans, which can hopefully avoid the usage of ritonavir. The drug target of PF-07321332, M^pro, plays a role in the viral polyprotein pp1a and pp1ab processing that is essential in the SARS-CoV-2 life cycle[8]. The M^pro gene has been observed to be relatively conserved among various SARS-CoV-2 variants; therefore, M^pro represents a promising target for drug development for SARS-CoV-2.

Other than PF-07321332, multiple series of SARS-CoV-2 M^pro inhibitors have been developed or discovered[6,9–23]. With a few exceptions[11,13,14,19,23], the majority of these molecules are peptidomimetics, which often exhibit poor pharmacokinetic (PK) properties. In this study, we report a further peptidomimetic M^pro inhibitor—RAY1216— which has been recently approved as an oral COVID-19 antiviral medicine in China, with the generic name 'leritrelvir'. The development of RAY1216 was inspired by the successful hepatitis C virus (HCV) protease inhibitor discovery programme reported for telaprevir[24–27]. RAY1216 features an α-ketoamide warhead and incorporates chemical

moieties known to confer selectivity towards coronavirus M^pro. Here we characterize in detail the kinetics of SARS-CoV-2 M^pro inhibition by RAY1216 and determine the crystal structure of the covalent adduct with SARS-CoV-2 M^pro. Furthermore, the antiviral activity, protection against SARS-CoV-2 variants in animal models and PK properties are reported, and compared with those of PF-07321332.

## Structure of RAY1216

RAY1216 (Fig. 1) was developed via multiple rounds of optimization conducted at P1, P2, P3 and P4 moieties, and finally, the covalent warhead was changed from a nitrile in PF-07321332 to an α-ketoamide moiety. The details of the structure–activity relationship optimizations will be further disclosed in a separate report. RAY1216 was chemically synthesized (Supplementary Fig. 1), and the identity of the product is confirmed using nuclear magnetic resonance (NMR) spectroscopy (Supplementary Figs. 2–4). The inhibitor features a cyclopentyl-substituted α-ketoamide warhead, a pyroglutamine with a pyrrolidinone side chain at P1 (this moiety is known to mimic glutamine, which dominates in the P1 position of coronavirus M^pro recognition sequences[28]), a P2 cyclopentylproline, a P3 cyclohexylglycine and a P4 tri-fluoroacetamide (Fig. 1). The absolute configuration of the synthesized RAY1216 was confirmed using X-ray crystallography (Supplementary Fig. 5). Inhibition assays show that RAY1216 has high specificity towards SARS-CoV-2 M^pro (Extended Data Table 1).

## In vitro inhibition of M^pro by RAY1216 compared with PF-07321332

We used a peptide cleavage assay based on fluorescence resonance energy transfer[29] to monitor SARS-CoV-2 M^pro activity (Supplementary Figs. 6 and 7), and we estimated a Michaelis constant ($K_M$) of 31 μM and a turnover number ($k_{cat}$) of 0.12 s$^{-1}$ for M^pro (Supplementary Figs. 6–8 and Supplementary Tables 2 and 3). To compare the inhibition by RAY1216 with that by PF-07321332, M^pro (final concentration 80 nM as determined by the Bradford assay) was added to a solution of substrate (20 μM) and inhibitor (maximum concentration 444 nM, 2:3 dilution series down to 17 nM) in the assay buffer. The increase in fluorescence intensity was monitored in real time over a period of 1 h. Representative replicates for RAY1216 or PF-07321332 are shown in Fig. 2 (Supplementary Figs. 8 and 9). Both compounds showed a gradual onset of inhibitory activity; an initial relatively uninhibited phase in product formation is followed by a gradual approach to pseudo-equilibrium ('slow binding' inhibition[30,31]). Compound concentrations substantially lower than the nominal enzyme concentration caused a prominent inhibitory effect ('tight binding' inhibition[32–34]). The time course of the assay in the absence of inhibitors ([$I$] = 0) was markedly nonlinear owing to substrate depletion. Under these particular experimental conditions, the classic algebraic method of enzyme kinetic analysis, based on the first-order apparent rate constant '$k_{obs}$'[35], cannot be used. Instead, combined progress curves obtained at various inhibitor concentrations were fit globally to a system of first-order ordinary differential equations (ODE) solved by the software package DynaFit[36,37].

The data versus model overlay plots in Fig. 2a,b illustrate that the overall inhibitory potencies of RAY1216 and PF-07321332 are very similar. Note that at the three highest inhibitor concentrations ([$I$] = 444, 296 and 198 nM), the reaction progress curves become nearly horizontal at the end of the assay in Fig. 2a,b. However, also note that the approach to the quasi steady state is markedly slower for RAY1216 than for PF-07321332. This fundamental difference between the two compounds is made most clearly visible in the instantaneous rate plots shown in Fig. 2c,d. For example, at the highest inhibitor concentration ([$I$] = 444 nM, bottom curve shown in red in Fig. 2c), it takes approximately 20 min for the enzyme to become fully inhibited by RAY1216. By contrast, it takes less than 1 min for the enzyme to become fully inhibited by PF-07321332 under identical conditions. Note in Fig. 2c,d that the reaction rate does not decrease to zero even at inhibitor

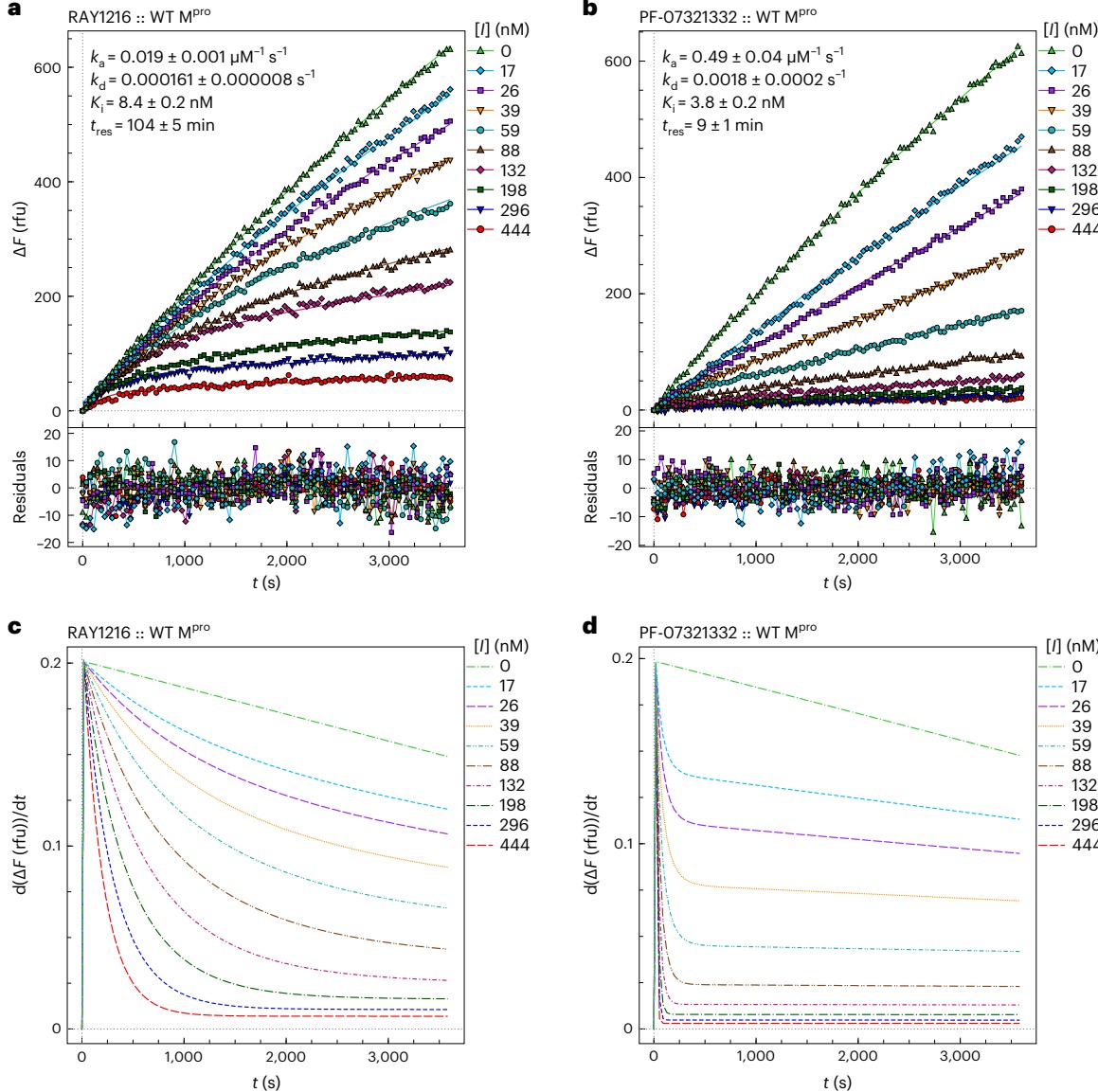

**Fig. 2 | SARS-CoV-2 M$^{pro}$ inhibition by RAY1216 and PF-07321332. a**, Progress curves of M$^{pro}$ inhibition (80 nM M$^{pro}$, 20 μM substrate) at different RAY1216 concentrations (data points); reactions were started without preincubation. Progress curves are fit in DynaFit[36,37] using the ODE method (smooth model curves), and residuals of the fits are shown. Δ$F$, fluorescence intensity change; rfu, relative fluorescence units. **b**, Progress curves of M$^{pro}$ inhibition by PF-07321332 under the same experimental conditions, and they are fit in DynaFit using the same analysis procedure. Inhibition parameters (mean ± s.d., $n$ = 3) determined from replicates are shown. **c,d**, Instantaneous reaction rates derived from the fits to the progress curves of M$^{pro}$ inhibition by RAY1216 (**c**) and PF-07321332 (**d**). See 'Enzyme kinetics' in Supplementary Information for details of data analysis procedures.

concentrations substantially higher than the enzyme concentration. This shows the effective kinetic reversibility of the observed enzyme–inhibitor interactions despite the fact that the crystal structure shows a covalent binding mode (see below). Thus, RAY1216 appears to be an example of a 'reversible covalent' inhibitor[38]. As the equilibrium dissociation constants $K_i = k_d/k_a$ (where $K_i$ is the inhibition constant, $k_d$ is the dissociation rate constant and $k_a$ is the association rate constant) for the two compounds are similar (Extended Data Table 2a), while it takes very much longer for RAY1216 to fully associate with the enzyme, it necessarily means that not only the association rate constant but also the dissociation rate constant is very much lower for RAY1216, compared with PF-07321332. In that sense, RAY1216 could be described as a 'slow-on, slow-off' inhibitor, whereas the PF-07321332 inhibition of M$^{pro}$ is 'fast on, fast off'.

The results of a comprehensive kinetic analysis using multiple replicates ($n$ = 3, for each inhibitor) are summarized in Extended Data

Table 2a (see Supplementary Tables 3 and 4 for detailed analysis). The $K_i$ and the drug-target residence time ($t_{res}$) were computed from these primary regression parameters using the usual formulas[39], while assuming that both inhibitors are kinetically competitive with the fluorogenic peptide substrate (see 'Enzyme kinetics' in Supplementary Information for details). The results summarized in Extended Data Table 2a indicate that RAY1216 has a more than an order of magnitude (12×) lower dissociation rate constant compared with PF-07321332. Thus, the drug-target $t_{res}$ for RAY1216 is measured in hours (1.7 h), instead of in minutes (9 min) in the case of PF-07321332. At the same time, the equilibrium binding affinity of RAY1216 (8.4 nM) measured by $K_i$ is only approximately twofold lower than that of PF-07321332. Note that $K_i$ = (3.8 ± 0.2) nM reported here for PF-07321332 is in good agreement with $K_i$ = 3.1 (1.5–6.8) nM previously reported by Pfizer[6]. The observed enzyme inhibition kinetics, in particular the drug-target $t_{res}$ results listed in Extended Data Table 2a, is consistent with slow–tight inhibition

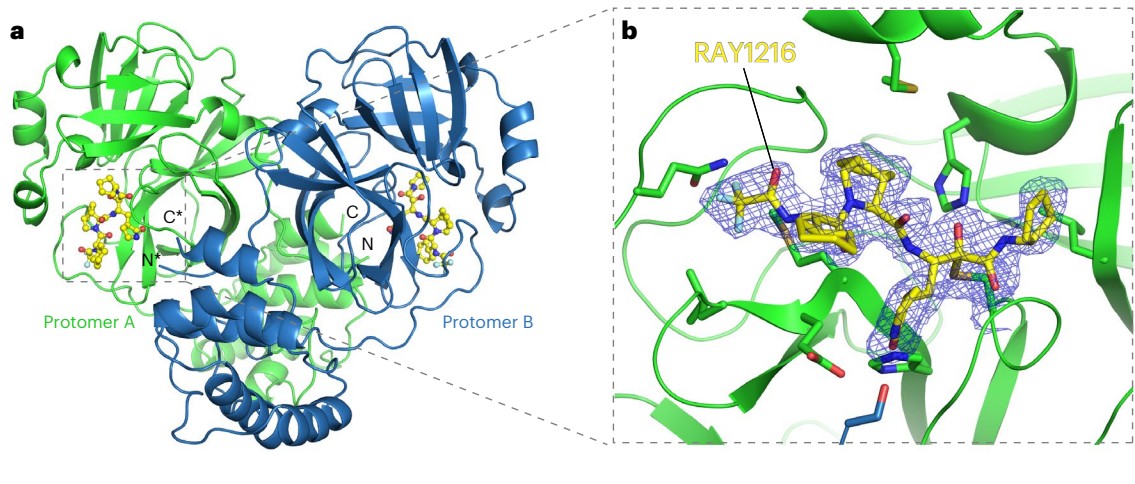

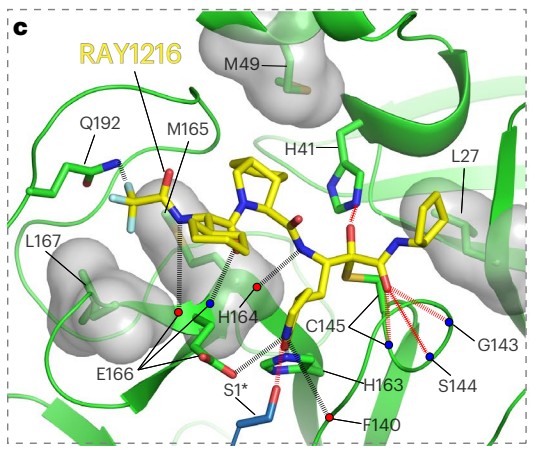

**Fig. 3 | Crystal structure of RAY1216 in complex with SARS-CoV-2 M^pro.**
**a**, Cartoon representation of the dimeric M^pro bound to RAY1216. Protomer A is in green, protomer B is in blue and RAY1216 is shown as yellow ball-and-stick models in active sites of both M^pro protomers. Asterisks mark structural features from protomer B. **b**, A zoom-in view of the RAY1216-bound active site of protomer A. An electron density (2Fo-Fc) map (blue mesh, contoured at 1.3$\sigma$) is shown around the bound RAY1216 and the catalytic Cys145 side chain (also see Supplementary Fig. 10 for omit map densities). Clear electron density is observed for the thiohemiketal bond formed between the bound RAY1216 α-keto carbon and the catalytic Cys145

sulfur. **c**, Same view as in **b** showing detailed interactions between RAY1216 and the active site of M^pro. Selected side chains of interacting residues are shown; backbone carbonyl and amide are represented as red and blue dots. **d**, Detailed interactions between PF-07321332 and the active site of M^pro (based on PDB 7RFW (ref. 6)) are shown in the same view as in **c**. In **c** and **d**, molecular surfaces of selected residues involved in hydrophobic contacts with bound inhibitors are shown. Hydrogen bonds are shown in dashed lines. Extra hydrogen bonds formed by RAY1216 to M^pro or hydrogen bonds of different properties to M^pro between RAY1216 and PF-07321332 are highlighted with colours.

of M^pro by RAY1216, suggesting that the enzyme–inhibitor complex (E–I) formed by RAY1216 is more stable than that formed by PF-07321332. We further performed differential scanning fluorimetry to show that the binding of RAY1216 increases the thermal denaturation midpoint temperature ($T_m$) of M^pro by 20 °C, while the binding of PF-07321332 increases M^pro $T_m$ by 11 °C (Extended Data Fig. 1a). These results confirm that RAY1216 forms a more stable enzyme–inhibitor complex.

## Structure of RAY1216 bound to SARS-CoV-2 M^pro

To further understand the activity of RAY1216, we soaked SARS-CoV-2 M^pro crystals with 6 mM RAY1216 in crystallization solution and the structure of RAY1216 bound to M^pro at 2.0 Å resolution was determined using X-ray diffraction (Fig. 3a and Supplementary Table 6). We identified unambiguous electron density consistent with RAY1216 molecules in both active sites of the M^pro dimer (Fig. 3a and Supplementary Fig. 10), and the dimer appears to be largely symmetric (Fig. 3a and Supplementary Fig. 11). The electron density shows that RAY1216 is covalently attached to M^pro via a thiohemiketal bond formed between the γ-sulfur of the catalytic Cys145 and the α-keto carbon of the RAY1216 warhead (Fig. 3b and Supplementary Fig. 10). The α-ketoamide warhead at the

inhibitor P1′ position is able to interact with the M^pro active site through a number of potential hydrogen bonds: the oxyanion (or hydroxyl) group of the thiohemiketal accepts a hydrogen bond from His41, and the warhead amide oxygen is within hydrogen bond accepting distance of the backbone amides of Gly143, Ser144 and Cys145, which form the canonical cysteine protease 'oxyanion hole' (Fig. 3c). These interactions are consistent with the proposal that the α-ketoamide represents a superior warhead through its ability to engage two hydrogen bonding interactions to the target protease catalytic centre, rather than just one[21], as seen for aldehyde[21,40] or Michael acceptor[21,41] warheads. The cyclopentyl substituent on the warhead amide is well defined by the electron density (Fig. 3b and Supplementary Fig. 10) and is situated 4.2 Å from the side chain of M^pro Leu27, showing a hydrophobic contact between the cyclopentyl moiety and the aliphatic Leu27 side chain (Fig. 3c).

In the P2 position of RAY1216, the peptide bond is stabilized within the cyclopentylproline moiety previously used at the P2 position of telaprevir[16,42]. Electron density shows that the hydrophobic cyclopentyl ring slots snugly into the groove between M49 and M165 (Fig. 3b,c and Supplementary Fig. 10). Plasticity has been observed for the S2

substrate binding pocket that accommodates the P2 moiety upon inhibitor binding (Supplementary Fig. 11)[43]. It has been shown that S2 pockets in coronavirus $M^{pro}$ have a strong preference towards hydrophobic amino acids, particularly leucine[28,44,45]. It has also been shown in a separate study that dimethylcyclopropylproline and cyclopentylproline, used in boceprevir and telaprevir, respectively (Fig. 1), when incorporated in α-ketoamide $M^{pro}$ inhibitors, can each occupy the S2 pocket with similar potencies[16].

The P3 moiety of RAY1216 features a cyclohexyl group that extends towards the exterior of the active site without making any direct contacts with $M^{pro}$ (Fig. 3c). The density for the cyclohexyl *para*-carbon positioned furthest from the active site cavity is weak (Fig. 3b), suggesting that the cyclohexyl group remains relatively flexible within the inhibitor–enzyme complex. Nevertheless, it has been reported that substituents at the P3 position can affect both drug potency and pharmacokinetic properties[6,16].

RAY1216 and PF-07321332 share the same γ-lactam and tri-fluoroacetamide moieties at P1 and P4, respectively. The P1 γ-lactam is known as an optimal fragment for viral protease inhibition as it mimics glutamine and has been proven to be responsible for potent inhibitory activity against a variety of enzymes with specificity towards native substrates with a P1 glutamine[6,16,46]. In the RAY1216:$M^{pro}$ complex, the γ-lactam nitrogen donates potential hydrogen bonds to the backbone carbonyl oxygen of Phe140 (3.19 Å), to the carboxylate of Glu166 (3.17 Å) and to the side chain hydroxyl of Ser1 from the second monomer of the $M^{pro}$ dimer (Fig. 3c). The γ-lactam carbonyl oxygen accepts a hydrogen bond (2.54 Å) from the imidazole of His163 (Fig. 3c). These interactions have also been observed in the complex formed between PF-07321332 and $M^{pro}$ (Fig. 3d)[6]. Clear electron density is observed for the P4 tri-fluoroacetamide capping moiety in the RAY1216:$M^{pro}$ complex structure (Fig. 3b and Supplementary Fig. 10). This moiety contacts the Leu167 side chain and accepts a hydrogen bond from Gln192 amide (Fig. 3c). Equivalent interactions have been observed in the PF-07321332:$M^{pro}$ complex structure (Fig. 3d)[6]. In summary, despite differences in the P1′ warhead, P2 bicycloproline and P3 substituent structures, interactions mediated by the P1 γ-lactam and P4 tri-fluoroacetamide moieties are largely maintained between RAY1216 and PF-07321332.

## Antiviral activities of RAY1216 in the cell culture and mouse model

Based on the encouraging in vitro activity of RAY1216, we next sought to investigate the inhibitory activity of RAY1216 towards SARS-CoV-2 infection in the cell and mouse model. The 50% cytotoxic concentration ($CC_{50}$) of RAY1216 was determined to be 511 μM for Vero E6 cells (Supplementary Fig. 12). In the virus plaque-reduction assays, the half-maximal effective concentration ($EC_{50}$) values for RAY1216 against different SARS-CoV-2 variants are 116 nM (wild type (WT)), 80 nM (Alpha), 88 nM (Beta), 69 nM (Delta), 81 nM (Omicron BA.1), 91 nM (Omicron BA.5) and 135 nM (Omicron XBB.1.9.1) (Fig. 4a); these values are comparable to but slightly less favourable than those of PF-07321332 (Fig. 4a, Extended Data Fig. 2, Supplementary Table 7 and Extended Data Fig. 3). The selectivity indices ($CC_{50}/EC_{50}$) of RAY1216 are 4,400 (WT), 6,390 (Alpha), 5,810 (Beta), 7,410 (Delta), 6,310 (Omicron BA.1), 5,620 (Omicron BA.5) and 3,790 (Omicron XBB.1.9.1).

We further characterized the protective effect of RAY1216 against virus infection in a human ACE2 (K18-hACE2) transgenic mouse model[47]. Mice were intranasally challenged with lethal doses ($10^5$ plaque forming units (PFU)) of SARS-CoV-2 (Delta variant), and the protective effect of RAY1216 was assessed. The mortality of the mice in the untreated virus-infected group was 100% at 5 days post-infection. RAY1216 administered at three different doses (600 mg $kg^{-1}$ $d^{-1}$, 300 mg $kg^{-1}$ $d^{-1}$ and 150 mg $kg^{-1}$ $d^{-1}$) was able to protect mice infected with SARS-CoV-2 by 100%, 43% and 14%, respectively (Fig. 4b). This result suggests that treatment with RAY1216 effectively prolonged survival of mice infected with SARS-CoV-2. To examine the effect of RAY1216 on lung virus titre and pathology, a separate set of experiments was performed with a non-lethal dose of virus inoculum ($10^{3.5}$ PFU). Mice treated with RAY1216 (600 mg $kg^{-1}$ $d^{-1}$ and 300 mg $kg^{-1}$ $d^{-1}$) had significantly decreased lung viral titres compared with the infection-only group (Fig. 4c–e). Compared with the infection-only group, the group treated with RAY1216 (600 mg $kg^{-1}$ $d^{-1}$) was able to reduce lung virus titre by more than 1 log unit. This effect may be slightly weaker for RAY1216 compared with PF-07321332 under the same experimental set-up (Fig. 4d,e), but the difference is not statistically significant. The lung histopathology of infected mice, compared with that of infected mice treated with RAY1216, shows that RAY1216 administered at 600 mg $kg^{-1}$ $d^{-1}$ and 300 mg $kg^{-1}$ $d^{-1}$ reduced virus-induced pathology (Fig. 4f). RAY1216 administered at a dose of 600 mg $kg^{-1}$ $d^{-1}$ provided a similar level of protection against lung tissue inflammation injury to that observed with PF-07321332 (Fig. 4f).

## Pharmacokinetics of RAY1216

Pharmacokinetics can substantially influence drug therapeutic efficacy. We examined the stability of RAY1216 in plasmas of different species (Extended Data Fig. 4). RAY1216 shows good stability in mouse and rat plasmas; more than 80% of RAY1216 is retained after 2 h of incubation, and only a very small fraction (2.6–4%) of the drug exhibits epimerization at the P1 stereocentre. In cynomolgus monkey and human plasmas, 60% and 70% of RAY1216 are retained after 2 h of incubation and P1 epimerization accounts for 13% and 11% of drug concentration loss. In beagle dog plasma, only 39% of RAY1216 is retained and P1 epimerization accounts for 52% of RAY1216 concentration loss after 2 h of incubation (Extended Data Table 3).

We next examined in vivo pharmacokinetics of RAY1216 and PF-07321332 in head-to-head experiments in mice and rats (Fig. 5). Following intravenous (i.v.) administration, RAY1216 has plasma clearance rates in the range of 7.2–10 ml $min^{-1}$ $kg^{-1}$ (compared with 35–54 ml $min^{-1}$ $kg^{-1}$ for PF-07321332) and elimination half-lives in the range of 3.3–4.8 h (compared with 0.3–1.2 h for PF-07321332) giving total drug exposure integrated over time (as represented by the area under the curve up to the last quantifiable time point, $AUC_{0–last}$) in the range of 3,400–7,000 h ng $ml^{-1}$ (compared with 940–1,500 h ng $ml^{-1}$ for PF-07321332). Following oral (p.o.) administration, RAY1216 has elimination half-lives ranging between 1.8 h and 3.5 h (compared with 0.7–1.1 h for PF-07321332) giving total drug exposure integrated over time ($AUC_{0–last}$) in the range of 5,300–9,200 h ng $ml^{-1}$ (compared with 810–2,200 h ng $ml^{-1}$ for PF-07321332) (Extended Data Table 4). These PK characteristics represent an improvement over PF-07321332, which is associated with faster plasma clearance

**Fig. 4 | Antiviral activities of RAY1216 in the cell culture and animal model.** **a**, Inhibition of SARS-CoV-2 wild-type ancestral strain and variants in cell culture. The antiviral effect of RAY1216 against SARS-CoV-2 virus infection was assessed using plaque-reduction assay (mean ± s.d., *n* = 3). Virus inhibition titres ($EC_{50}$) are estimated from dose–response curves of percentage plaque-reduction versus RAY1216 concentration. **b**, Protection of K18-hACE2 transgenic mice from lethal SARS-CoV-2 challenge by RAY1216. **c**, Body weight change (mean ± s.d.) of K18-hACE2 transgenic mice infected with SARS-CoV-2 after receiving indicated daily oral doses of RAY1216, PF-07321332 or PBS control (*n* = 7). **d,e**, SARS-CoV-2 virus titres (mean ± s.d.) in mouse lung tissues at 3 days post-infection (**d**, *n* = 3) and 5 days post-infection (**e**, *n* = 7) after receiving the indicated daily doses of RAY1216 or PF-07321332. *$P \leq 0.05$; **$P \leq 0.01$; ***$P \leq 0.001$; NS, not significant as determined using one-way ANOVA with Tukey's HSD test compared with the virus group. **f**, Comparison of virus-induced histology changes in mouse lung tissues after receiving the indicated oral daily doses of RAY1216 or PF-07321332 (*n* = 3). Histology examples of no virus (NC) and virus (Virus) controls are included for comparison.

and shorter elimination half-lives under equivalent conditions in mouse and rat models.

We performed further PK experiments in K18-hACE2 transgenic mice under the same dosing condition (600 mg kg$^{-1}$ d$^{-1}$, p.o.)

used for antiviral animal experiments (Supplementary Fig. 13). Under this condition, RAY1216 gives a total drug exposure integrated over time (AUC$_{0-last}$) of 140,000 h ng ml$^{-1}$, which is ~6 times that of PF-07321332 (23,000 h ng ml$^{-1}$) (Supplementary Table 8).

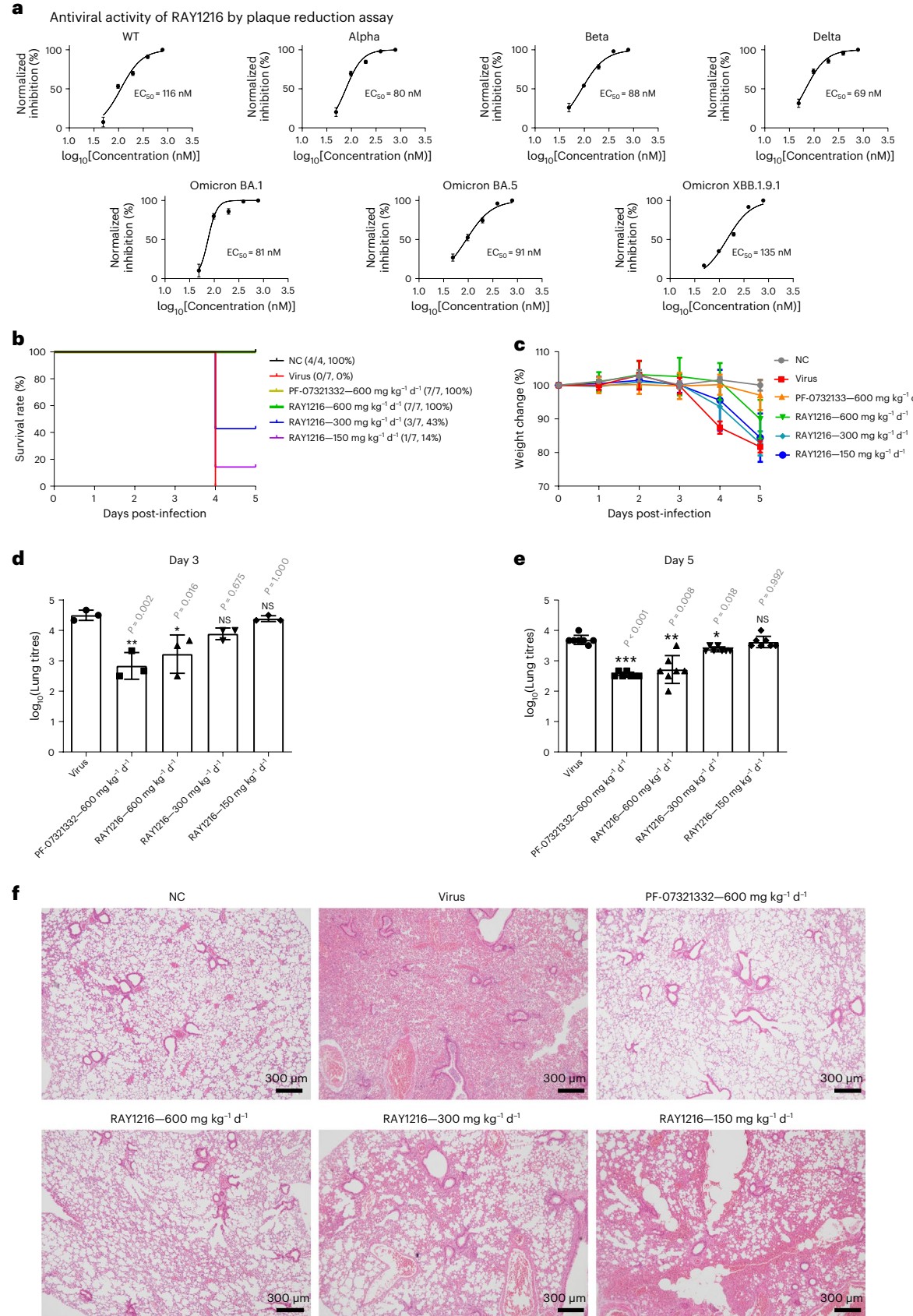

**a** Antiviral activity of RAY1216 by plaque reduction assay

WT — EC$_{50}$ = 116 nM
Alpha — EC$_{50}$ = 80 nM
Beta — EC$_{50}$ = 88 nM
Delta — EC$_{50}$ = 69 nM
Omicron BA.1 — EC$_{50}$ = 81 nM
Omicron BA.5 — EC$_{50}$ = 91 nM
Omicron XBB.1.9.1 — EC$_{50}$ = 135 nM

**b** Survival rate (%) vs Days post-infection
- NC (4/4, 100%)
- Virus (0/7, 0%)
- PF-07321332—600 mg kg$^{-1}$ d$^{-1}$ (7/7, 100%)
- RAY1216—600 mg kg$^{-1}$ d$^{-1}$ (7/7, 100%)
- RAY1216—300 mg kg$^{-1}$ d$^{-1}$ (3/7, 43%)
- RAY1216—150 mg kg$^{-1}$ d$^{-1}$ (1/7, 14%)

**c** Weight change (%) vs Days post-infection
- NC
- Virus
- PF-0732133—600 mg kg$^{-1}$ d$^{-1}$
- RAY1216—600 mg kg$^{-1}$ d$^{-1}$
- RAY1216—300 mg kg$^{-1}$ d$^{-1}$
- RAY1216—150 mg kg$^{-1}$ d$^{-1}$

**d** Day 3 — log$_{10}$(Lung titres)
- Virus
- PF-07321332—600 mg kg$^{-1}$ d$^{-1}$ (P = 0.002, **)
- RAY1216—600 mg kg$^{-1}$ d$^{-1}$ (P = 0.016, *)
- RAY1216—300 mg kg$^{-1}$ d$^{-1}$ (P = 0.675, NS)
- RAY1216—150 mg kg$^{-1}$ d$^{-1}$ (P = 1.000, NS)

**e** Day 5 — log$_{10}$(Lung titres)
- Virus
- PF-07321332—600 mg kg$^{-1}$ d$^{-1}$ (P < 0.001, ***)
- RAY1216—600 mg kg$^{-1}$ d$^{-1}$ (P = 0.008, **)
- RAY1216—300 mg kg$^{-1}$ d$^{-1}$ (P = 0.018, *)
- RAY1216—150 mg kg$^{-1}$ d$^{-1}$ (P = 0.992, NS)

**f** NC | Virus | PF-07321332—600 mg kg$^{-1}$ d$^{-1}$ | RAY1216—600 mg kg$^{-1}$ d$^{-1}$ | RAY1216—300 mg kg$^{-1}$ d$^{-1}$ | RAY1216—150 mg kg$^{-1}$ d$^{-1}$ (300 μm scale bars)

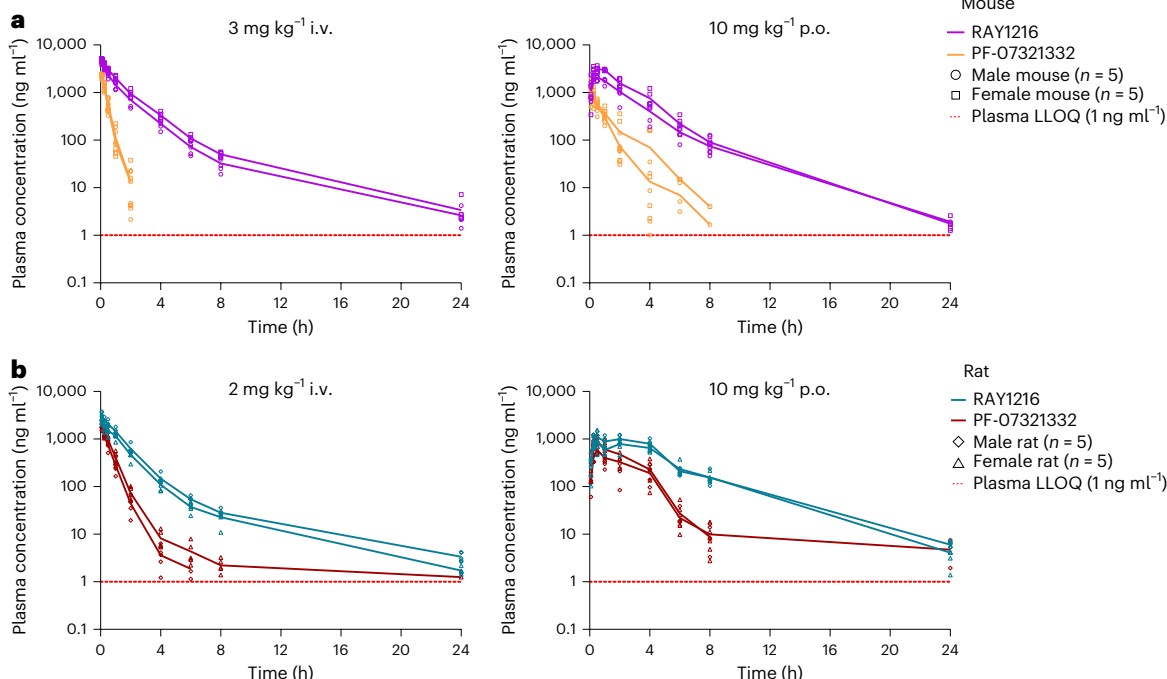

**Fig. 5 | Plasma concentrations of RAY1216 and PF-07321332 after i.v. injection dosing and gavage (p.o.) dosing in mouse and rat models. a**, The pharmacokinetics in the mouse models. **b**, The pharmacokinetics in the rat models. In each dosing experiment, male (circles and diamonds) and female (squares and triangles) animal groups were tested. Each group consisted of five animals, and the data of each animal are shown with group average values connected by lines. The red dashed lines represent the lower limit of quantitation (LLOQ, 1 ng ml⁻¹) of plasma drug concentration.

Under this dosing condition, RAY1216 can maintain a plasma concentration above $EC_{90}$ for at least 8 h. By contrast, PF-07321332 can maintain only a plasma concentration above $EC_{90}$ for 4–5 h (Supplementary Fig. 13). The animal PK data indicate that RAY1216 may have a promising human PK profile.

## M^pro mutants and RAY1216 inhibition

Since the emergence of Omicron variants, circulating SARS-CoV-2 strains have been carrying the P132H mutation[48]. We purified the P132H M^pro, and this enzyme was found to have a $K_M$ of 35 µM and a $k_{cat}$ of 0.18 s⁻¹ similar to those of WT M^pro ($K_M$ = 31 µM, $k_{cat}$ = 0.12 s⁻¹) (Supplementary Fig. 7 and Supplementary Tables 2 and 5). In enzyme inhibition assays involving P132H M^pro (Extended Data Fig. 5 and Supplementary Table 5), RAY1216 has a $K_i$ of 8.4 nM and a $t_{res}$ of 76 min, while PF-07321332 has a $K_i$ of 4.9 nM and a $t_{res}$ of 3 min. Differential scanning fluorimetry experiments show that the binding of RAY1216 and PF-07321332 increases P132H M^pro $T_m$ by 20 °C and 10 °C, respectively (Extended Data Fig. 1b). Therefore, compared with WT M^pro inhibition, RAY1216 and PF-07321332 retain very similar enzyme inhibition characteristics towards P132H M^pro, consistent with a previous report on PF-07321332 (ref. 49) and consistent with the fact that both drugs retain a good antiviral effect towards Omicron variants.

It has been reported that the use of PF-07321332 can induce M^pro mutations in both laboratory and clinical settings, although these mutations are yet to be widely found in circulating viruses[50]. Using a SARS-CoV-2 replicon system[51], we investigated the inhibition of replicon-driven luciferase expression in human embryonic kidney (HEK) 293T cells by RAY1216 and PF-07321332 (Fig. 6a). RAY1216 and PF-07321332 are able to inhibit the replicon encoding WT M^pro with $EC_{50}$ values of 50 nM and 33 nM. We further tested effects of selected M^pro single mutants[50] G15S, M49L, F140L and ΔP168, and selected double mutants[52] L50F/E166V and E166A/L167F, on the inhibition of both drugs. The single mutants reduce RAY1216 and PF-07321332 inhibition 2.4–12.8-fold, and each mutation affects both drugs to similar degrees

(Extended Data Table 5). The double mutants have a weaker effect on RAY1216 inhibition, reducing inhibition 4.7–6.5-fold. For comparison, E166A/L167F reduces PF-07321332 inhibition 16.4-fold and L50F/E166V reduces PF-07321332 inhibition at least 150-fold (Extended Data Table 5). Among these mutations, positions 49, 140, 166 and 167 are in direct contact with RAY1216 by either hydrophobic interaction or hydrogen bonding. The other tested mutation sites are in the direct vicinity of the M^pro catalytic pocket where both RAY1216 and PF-07321332 bind (Fig. 6b).

## Discussion

In this study, we characterize the inhibition of SARS-CoV-2 M^pro by RAY1216, an optimized peptidomimetic inhibitor. We find that, probably due to more stable interaction with M^pro, RAY1216 possesses superior drug-target residence time, when compared with PF-07321332 (nirmatrelvir), the active antiviral component in Paxlovid. It has recently emerged that drug-target residence time is an important parameter to optimize for drug efficacy[39,53,54]. We further report that RAY1216 has better pharmacokinetic properties compared with PF-07321332. Improved pharmacokinetic properties may allow RAY1216 to be used without ritonavir, which is known to have significant unwanted drug–drug interactions. However, PF-07321332 is slightly favoured over RAY1216 in reducing mouse lung viral titre. These results suggest that the drug-target residence time alone, as determined in biochemical kinetic assays, cannot solely dictate pharmacological efficacy. RAY1216 has completed its phase III clinical trial as a single-component drug[55], and it has been approved by the National Medical Products Administration of China for COVID-19 treatment with the generic drug name 'leritrelvir'. Despite its successful entry into clinical use, our results show that RAY1216, as with all antiviral drugs, is facing significant challenges posed by drug resistance mutations. Further investigations of resistance mechanisms, in particular those that are specific to the M^pro enzyme, should help the development of future M^pro inhibitors as therapeutic agents.

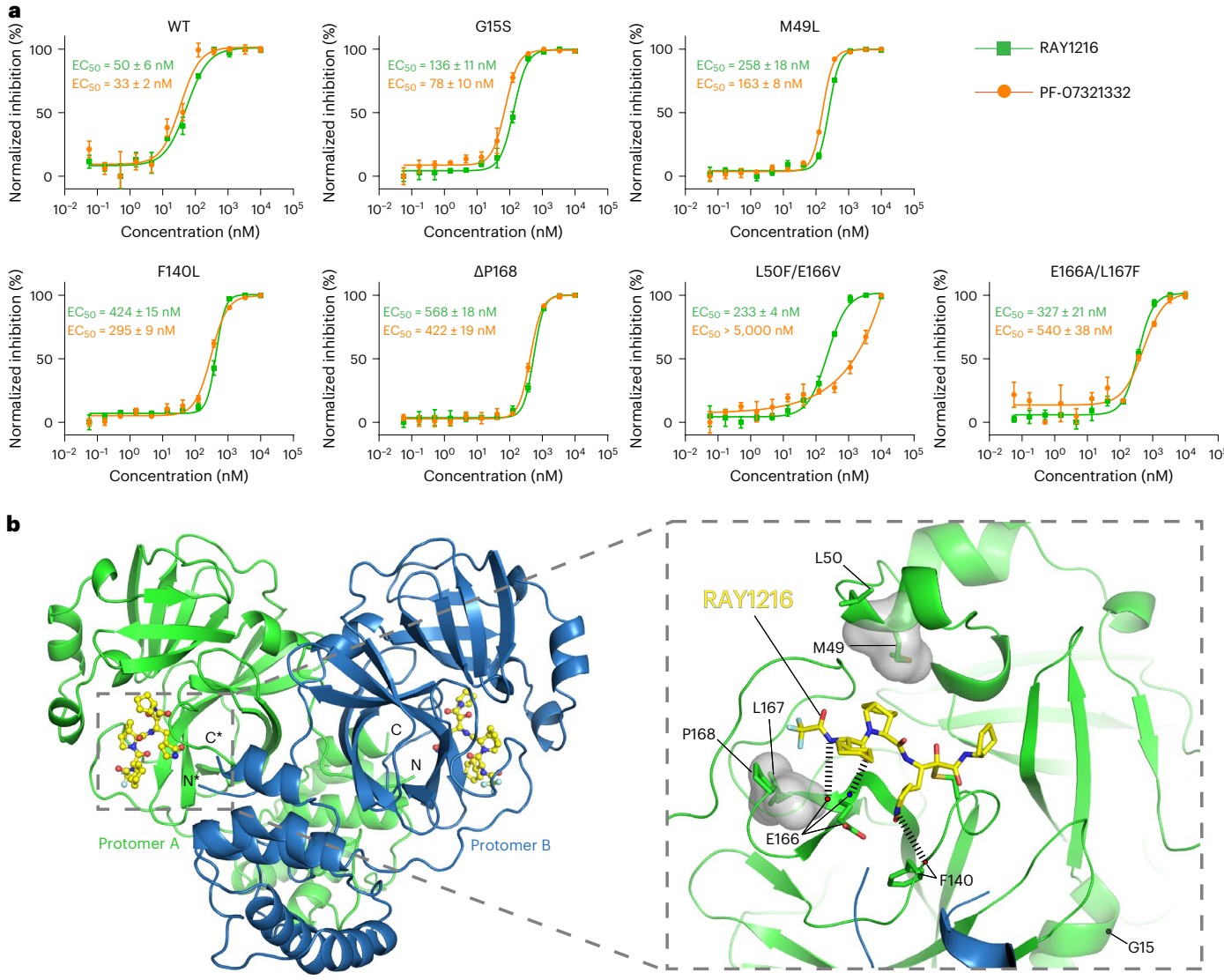

**Fig. 6 | Inhibition of SARS-CoV-2 replicons in 293T cells. a**, Dose–response curves of replicon inhibition by RAY1216 and PF-07321332. Percentage inhibition (mean ± s.d., $n$ = 3) of replicons bearing indicated M$^{pro}$ mutations was assessed by measuring replicon-expressed luciferase activity. **b**, Tested mutation positions mapped onto the RAY1216:M$^{pro}$ inhibition complex structure. Hydrogen bonds between RAY1216 and mutated residues are indicated by dashed lines; transparent surfaces mark residues engaging hydrophobic contact to RAY1216.

## Methods

### Ethics statement

The antiviral studies were performed in the Guangzhou Customs District Technology Center Biosafety Level 3 (BSL-3) Laboratory and approved by the Guangzhou Medical University Ethics Committee of Animal Experiments (Institutional Animal Care and Use Committee (IACUC) certificate number GZL0008). The pharmacokinetic studies were approved by the IACUC of Precedo Pharmaceuticals. The IACUC numbers for rat and mouse (including Institute of Cancer Research (ICR) mouse and K18-hACE2 mouse) pharmacokinetic studies are IACUC-20230303-2 and IACUC-20230303-3, respectively. All plasmas used in the RAY1216 plasma stability experiment are from commercial sources. All experiments complied with the relevant ethical regulations.

### Statistical analysis

The sample sizes were predetermined according to previous studies[16,47]. The experiments were not randomized, and investigators were not blinded to allocation during experiments and outcome assessments.

### Chemical synthesis and characterization of RAY1216

RAY1216 was synthesized through an 11-step process. The synthesized RAY1216 was characterized using NMR spectroscopy, infrared spectroscopy and mass spectrometry. The absolute configuration of the synthesized RAY1216 was confirmed using X-ray crystallography. Details of RAY1216 chemical synthesis and characterization are described in Supplementary Information.

### Recombinant protein production

Based on a previous study[21], a construct encoding SARS-CoV-2 M$^{pro}$ (*ORF1ab* 3264-3569, GenBank code MN908947.3) was subcloned into the pGEX-6p-1 vector between the BamHI and XhoI restriction sites with extra C-terminal extension encoding a human rhinovirus 3C enzyme recognition site linked to a 10×His tag. The resulting wild-type and P132H mutant construct was verified by DNA sequencing. The construct plasmid was transformed into BL21 (DE3) *Escherichia coli* cells (Vazyme, number C504-02/03), and scale-up expression (~6 l) was started from a single colony in lysogeny broth medium supplemented with 100 μg ml$^{-1}$ of ampicillin at 37 °C. The cells were induced with

0.5 mM isopropyl β-D-1-thiogalactopyranoside when the optical density at 600 nm reached 0.8. Cells were allowed to grow post-induction for 20 h at 16 °C.

The cells were collected through centrifugation, and the cell pellet was lysed in the lysis buffer (20 mM Tris, pH 7.8, 150 mM NaCl, 10 mM imidazole) through sonication on ice. The cell lysate was cleared using high-speed centrifugation (20,000 × $g$ at 4 °C for 1 h). The supernatant was mixed with Ni-NTA resin for ~2 h at 4 °C on a shaker. The Ni-NTA resin was washed with two buffers of different imidazole concentrations (20 mM Tris, pH 7.8, 150 mM NaCl, 20 mM/50 mM imidazole) each for over 30 resin volumes to remove contaminants. The target protein was eluted by the elution buffer (20 mM Tris, pH 7.8, 150 mM NaCl, 500 mM imidazole). Human rhinovirus 3C enzyme, 400 U, was added into the eluted protein to remove the C-terminal histidine tag, and the mixture was dialyzed at 4 °C overnight in dialysis buffer (20 mM Tris, pH 7.8, 150 mM NaCl, 1 mM DTT) using a dialysis bag with a molecular weight cut-off of 10 kDa. The dialyzed mixture was reloaded onto the Ni-NTA resin, and His-tag-free target protein was collected from the flow-through.

As the expressed $M^{pro}$ contains the native $M^{pro}$ cleavage sequence 'SAVLQ/SGFRK' found between Nsp4 and Nsp5 ($M^{pro}$) in the SARS-CoV-2 Nsp polyprotein (the slash indicates the $M^{pro}$ cleavage site) near the N-terminus, $M^{pro}$ auto-cleaving activity generates an authentic N-terminus during protein expression. Purified $M^{pro}$ was concentrated by a 10 kDa molecular weight cut-off Amicon Ultra 50 centrifugal filters (Merck Millipore) at 4 °C to ~10 mg ml$^{-1}$. Concentrated protein was either used for crystallization without freezing or flash frozen in liquid nitrogen and stored under −80 °C.

### Enzyme kinetic assay and analysis

The enzyme assays were performed in enzyme kinetic buffer (20 mM Tris pH 7.8, 150 mM NaCl, 1 mM DTT and 100 µg ml$^{-1}$ bovine serum albumin) using Dabcyl-KTSAVLQ/SGFRKME-Edans (Beyotime, number P9733; '/' indicates the $M^{pro}$ cleavage site) as the substrate. The reactions were carried out in 96-well black flat-bottom plates with a final reaction volume of 200 µl. A 10 s plate-shaking procedure was used before the data collection. The fluorescent signal by enzyme cleavage of the substrate was monitored on a Molecular Devices FlexStation 3 reader with filters for excitation at 340 nm and emission at 490 nm at 20 °C. The data were recorded using SoftMax Pro 7.0 Software. Minimal data point collection interval time was set to collect as many data points as possible. Enzyme kinetic assay results were analysed using the software package DynaFit[36,37] using methods detailed in 'Enzyme kinetics' in Supplementary Information.

### Thermal stability assay

$M^{pro}$ protein, 6 µM, was preincubated with 12 µM PF-007321332 or RAY1216 in reaction buffer PBS (containing 0.1% DMSO) at 16 °C for 30 min. The apo $M^{pro}$ control experiment was performed with 6 µM $M^{pro}$ protein in PBS (containing 0.1% DMSO). The SYPRO orange dye (Sigma, number 5692) was added to a final concentration of 5×. The final reaction volume was 20 µl. The fluorescence of the tube was monitored under a temperature gradient range from 25 °C to 94 °C with 0.5 °C min$^{-1}$ incremental step in a real-time PCR machine. The raw fluorescence data were normalized to the highest value in each melting curve, and the normalized fluorescence curves were fitted to a Boltzmann equation to estimate the melting temperature $T_m$ (the midpoint of the unfolding transition) in GraphPad Prism 8.0.

### $M^{pro}$ crystallization and crystal soaking

Apo $M^{pro}$ crystals were crystallized by mixing 1 µl of freshly purified $M^{pro}$ (without freezing) at 10 mg ml$^{-1}$ with 1 µl crystallization solution (0.1 M MES monohydrate pH 6.5, 12% w/v polyethylene glycol 20,000) using the hanging drop vapor diffusion method at 16 °C. Crystals normally grew overnight. The apo crystals were flash frozen in cryoprotection

solution (0.1 M MES monohydrate pH 6.5, 12% w/v polyethylene glycol 20,000, 40% glycerol) using liquid nitrogen. To obtain RAY1216 soaked crystals, apo crystals were transferred into the crystallization solution supplemented with 6.6 mM RAY1216 and 3% DMSO (from the RAY1216 solution). The crystals were soaked for ~10 min at 16 °C. Finally, the crystals were briefly soaked in cryoprotection solution (0.1 M MES monohydrate pH 6.5, 12% w/v polyethylene glycol 20,000, 40% glycerol) supplemented with 6.6 mM RAY1216 before being frozen in liquid nitrogen.

### Data collection and structure determination

Single-crystal X-ray diffraction data were collected on beamline BL19U1 at the Shanghai Synchrotron Radiation Facility at 100 K using an Eiger X 16M hybrid-photon-counting detector. Data integration and scaling were performed using the XDS software (BUILT 20220220)[56]. Structures were determined by molecular replacement using the Phaser MR 2.8.3 (ref. [57]) programme in CCP4 7.1.018 (ref. [58]), with a SARS-CoV-2 $M^{pro}$ structure[7] (Protein Data Bank (PDB) code 7VH8) as the search model. Iterative manual model building was carried out in Coot 0.9.6 (ref. [59]). Final structures were refined with Refmac 5.8.0267 (ref. [60]). The data collection and structure refinement statistics are summarized in Supplementary Table 6.

### Cell lines and virus strains

African green monkey kidney epithelial (Vero E6) cells and HEK293T cells were purchased from the American Type Culture Collection and cultured in Dulbecco's modified Eagle's medium (DMEM, Gibco) supplemented with 10% fetal bovine serum (FBS; Gibco), 100 µg ml$^{-1}$ streptomycin (Gibco) and 100 U ml$^{-1}$ penicillin (Gibco). SARS-CoV-2 and its variants, namely, Alpha (B.1.1.7), Beta (B.1.351), Delta (B.1.617.2), Omicron BA.1 (B.1.1.529), Omicron BA.5 (BA.5.2) and Omicron XBB.1.9.1, were isolated from clinical samples and were deposited at the First Affiliated Hospital of Guangzhou Medical University. Viruses were propagated as previously described[61], the viruses were aliquoted and stored at −80 °C, and the titres of cultured viruses were estimated as 50% tissue culture infective doses (TCID$_{50}$) using the Reed–Muench method.

### Cytotoxicity and cytopathic effect inhibition assays

The CC$_{50}$ for RAY1216 in Vero E6 cells was determined using the MTT (3-(4,5-dimethylthazolk-2-yl)-2,5-diphenyl tetrazolium bromide) assay[62]. Different dilutions of RAY1216 and PF-07321332 were incubated with Vero E6 cells (5 × 10$^4$ cells per well) in 96-well plates for the cytotoxicity assay, and the concentrations of RAY1216 and PF-07321332 causing 50% cell death were determined as the CC$_{50}$ value. The 50% inhibition concentration (EC$_{50}$) of the virus-induced cytopathic effect (CPE) was used to investigate the efficacy of RAY1216 and PF-07321332 against SARS-CoV-2. A monolayer of Vero E6 cells was inoculated with 100 TCID$_{50}$ of SARS-CoV-2 wild type or variant strain at 37 °C for 2 h. After removal of the inoculum, the cells were incubated with DMEM containing different concentrations of RAY1216 or PF-07321332, 2% FBS and 2 µM P-glycoprotein (P-gp) inhibitor, CP-100356. Infected cells were observed under a microscope after 72 h of incubation to assess the CPE. Dose–response curves were plotted as CPE versus log(inhibitor concentrations). The EC$_{50}$ and EC$_{90}$ were estimated using regression analysis in IBM SPSS Statistics software version 25.0. The selectivity indices were determined using the ratio of CC$_{50}$ to EC$_{50}$.

### Plaque-reduction assay

Vero E6 cells at a density of 2 × 10$^5$ cells per well were incubated overnight as a monolayer of cells in 12-well plates. After being rinsed with PBS, the cells were incubated with 100 TCID$_{50}$ of SARS-CoV-2 wild type or variant strains for 2 h. The inoculum was removed, and the cells were overlaid with 1 ml of 0.8% agar formulated in DMEM containing different concentrations of RAY1216 or PF-07321332, 2% FBS and 2 µM

P-gp inhibitor, CP-100356. Plaque-reduction assay was performed in triplicates. The plates were placed upside down in 37 °C for 72 h of incubation. The plates were fixed with 4% formalin for 30 min. The overlays were then removed and stained with 0.1% crystal violet for 3 min. The plaques were visualized and counted. The percentage inhibition at each indicated drug concentration was normalized against the virus control, and dose–response curves were plotted as percentage inhibition versus log(inhibitor concentrations). The $EC_{50}$ and $EC_{90}$ were estimated using regression analysis in IBM SPSS Statistics software version 25.0.

## Virus inhibition assay by quantitative PCR

A total of $2 \times 10^5$ Vero E6 cells per well were seeded in 12-well plates in DMEM supplemented with 10% FBS and cultured for 24 h at 37 °C. The cells were washed twice with PBS before the addition of 500 μl inoculum containing 100 $TCID_{50}$ SARS-CoV-2 virus. The control wells were set up with DMEM containing 2% FBS. After incubation at 37 °C for 2 h, the inoculum was removed. The cells were incubated with DMEM containing different concentrations of RAY1216 or PF-07321332, 2% FBS and 2 μM P-gp inhibitor, CP-100356. The virus control wells were replaced with medium containing 2% FBS and 2 μM CP-100356, but without inhibitor. The supernatants were collected after 48 h for real-time quantitative PCR assay to assess viral gene expression. The viral RNA was extracted and reverse transcribed to obtain cDNA as the quantitative PCR template. quantitative PCR was performed using a probe specific to the SARS-CoV-2 *ORF1ab-N* gene (TIANDZ). Amplification was carried out on an ABI PRISM 7500 real-time PCR System (Applied Biosystems) using PCR cycles: 95 °C, 5 min, and 45 cycles of 95 °C, 10 s; 60 °C, 60 s; and 72 °C 30 s. The viral gene expression levels from different wells were normalized to the control wells as fold changes. Statistical significance between different drug concentrations was assessed using one-way analysis of variance (ANOVA) with Tukey's honest significant difference (HSD) test in IBM SPSS statistics software version 25.0.

## Antiviral and anti-inflammatory activity of RAY1216 in the mouse model

All antiviral experiments using animals passed the ethical review and were performed in strict accordance with the National Research Council Criteria and the Chinese Animal Protection Act. Female human ACE2 transgenic C57BL/6 (K18-hACE2 transgenic) mice[47,63], 5 to 6 weeks old and weighing 18–22 g, were acquired from GemPharmatech and housed under specific pathogen-free (SPF) conditions at the Guangzhou Customs District Technology Center Biosafety Level 3 (BSL-3) Laboratory, and the housing environment had controlled temperature (20–26 °C), humidity (40–70%) and lighting conditions (12 h light and 12 h dark cycles). The animals were fed every day with fodder purchased from the Beijing Keao Xieli Feed, and the general quality standards, hygienic standards and conventional nutritional ingredient index requirements in feeds were tested in accordance with GB14924.2-2001 and GB14924.3-2010 standards. The mice were randomly divided into six groups ($n = 7$): the control group, the group infected with SARS-CoV-2 virus (Delta variant (B.1.617.2)), treatment groups of three different RAY1216 concentrations (600 mg kg$^{-1}$ d$^{-1}$, 300 mg kg$^{-1}$ d$^{-1}$, 150 mg kg$^{-1}$ d$^{-1}$), and a PF-07321332 treatment group (600 mg kg$^{-1}$ d$^{-1}$). Mice were anesthetized by inhalation of 5% isoflurane, and each mouse was inoculated with 50 μl PBS containing a lethal dose of $10^5$ PFU SARS-CoV-2 (Delta variant) for the infected groups. For the control group, 50 μl PBS was administered intranasally. Then, 2 h after infection, the infected mice were intragastrically administered with RAY1216 (600 mg kg$^{-1}$ d$^{-1}$, 300 mg kg$^{-1}$ d$^{-1}$, 150 mg kg$^{-1}$ d$^{-1}$), PF-07321332 (600 mg kg$^{-1}$ d$^{-1}$) or PBS daily for 5 days. The weight change and mortality of the mice in each group were recorded daily for 5 days. To measure lung virus titres and to examine lung pathology, a separate set of experiments was performed under the same grouping and conditions except that each mouse was inoculated with a non-lethal dose of $10^{3.5}$ PFU SARS-CoV-2 (Delta variant) for the infected groups. At 3 days and 5 days post-infection, the

mice were killed, and lung tissues were collected to measure virus titres and to examine lung pathology.

## Stability of RAY1216 and epimerization of RAY1216

To understand the epimerization of RAY1216 at the P1 stereocentre, RAY1216-E (R-epimer of RAY1216 at P1) was synthesized using the described method for RAY1216 (Supplementary Fig. 1), except that 'methyl-(*R*)-2-((*tert*-butoxycarbonyl)amino)-3-((*S*)-2-oxopyrrolidin-3-yl) propanoate' was used as the starting material instead of RAY1216-1 to generate the P1 R-epimer of RAY1216. A 100 μM DMSO solution of RAY1216-E was prepared as the standard for liquid chromatography with tandem mass spectrometry (LC–MS/MS) analysis.

The stability of RAY1216 was evaluated in plasmas of five species. CD-1 mouse plasma and Sprague Dawley (SD) rat plasma were purchased from Vital River Laboratories. Cynomolgus monkey plasma was purchased from Xishan Zhongke Laboratory Animal. Beagle dog (#CAN00PLK2Y2N) and human (#HUMANPLK2P2N) plasmas were purchased from BioIVT. Pooled frozen plasma was thawed in a water bath at 37 °C before clots were removed by centrifugation. Then, 2 μl of 100 μM RAY1216 DMSO solution was added to 98 μl plasma in 96-well plates. Reaction plates were incubated at 37 °C, after 0 min, 10 min, 30 min, 60 min and 120 min of incubation; 300 μl stop solution (acetonitrile containing 100 ng ml$^{-1}$ of six internal standards: dexamethasone, glyburide, tolbutamide, verapamil, labetalol, celecoxib) was added to each well with mixing to precipitate protein that was removed by centrifugation. The RAY1216-E sample was prepared in the same way without incubation (0 min) to be used as the elution time and concentration analysis standard in LC–MS/MS. Subsequently, 10 μl of the supernatant was analysed on an LC–MS/MS system (AB Sciex API 4000 + UPCC) attached to a Chiralpak OX3R column (100 mm, 4.6 mm, 3 μm). The concentration of RAY1216 or RAY1216-E was estimated using peak area ratio normalized to that of RAY1216 or RAY1216-E at 0 min. Half-life was calculated from the elimination rate constant estimated from a plot of Ln %Remaining/100 versus incubation time.

## Animals in preclinical pharmacokinetic studies

The ICR mice and SD rats of SPF grade were purchased from Vital River Laboratories. All animal care and experimental procedures in pharmacokinetic studies were implemented in accordance with approved guidelines. All the animals were fed every day with the fodder purchased from Wuhan WQJX Bio-Technology and housed in controlled temperature (20–26 °C), humidity (40–70%) and lighting (12 h light and 12 h dark cycles) conditions. The PK parameters were calculated using Phoenix WinNonlin software (version 8.2.0).

## Mouse pharmacokinetics

Pharmacokinetic properties of RAY1216 and PF-07321332 following a single dose of 3 mg kg$^{-1}$ i.v. and a single dose of 10 mg kg$^{-1}$ p.o. were examined in ICR mice. Five male mice and five female mice were included in the experimental group for each administration mode and each compound. The mice in the p.o. group were fasted overnight before administration; the p.o. formulation contained '30% PEG400, 10% Solutol HS15, 2% Tween-80, 58% water', and the mice were fed 4 h post-dose. Blood samples were taken from the cheek at 0.083, 0.25, 0.5, 1, 2, 4, 6, 8 and 24 h after dosing and collected into tubes containing sodium heparin. Plasma samples were obtained through centrifugation (6,000 × *g* at 4 °C for 3 min) and were stored frozen at −80 °C before LC–MS/MS analysis.

## Rat pharmacokinetics

The pharmacokinetic properties of RAY1216 and PF-07321332 following a single dose of 2 mg kg$^{-1}$ i.v. and a single dose of 10 mg kg$^{-1}$ p.o. were examined in SD rats. A total of five male rats and five female rats were included in the experimental group for each administration mode and each compound. The rats in the p.o. group were fasted overnight

before administration; the p.o. formulation was the same as that of the mice, and the rats were fed 4 h post-dose. Blood samples were taken via jugular vein cannula at 0.083, 0.25, 0.5, 1, 2, 4, 6, 8 and 24 h after dosing and collected into tubes containing sodium heparin. Plasma samples were obtained and stored in the same manner as described in the mouse section above.

## K18-hACE2 mouse pharmacokinetics

To evaluate the pharmacokinetics of RAY1216 and PF-07321332 in the K18-hACE2 mouse, RAY1216 and PF-07321332 groups were set up and each group had five SPF-grade female 6-week-old K18-hACE2 mice (purchased from GemPharmatech). Each mouse was administrated with a single dose of 600 mg kg$^{-1}$ RAY1216 or PF-07321332 p.o. daily for 5 days (the same dose used in the antiviral animal experiment). On the fifth day, blood samples were taken from the cheek at 0.083, 0.25, 0.5, 1, 2, 4, 6, 8 and 24 h. The blood samples were processed and analysed with the same protocol as the ICR mouse samples.

## LC−MS/MS analysis of mouse plasma samples

For the mouse pharmacokinetic studies, 20 µl plasma samples (the blank sample used 20 µl blank plasma) were mixed with 200 µl of 50% methanol acetonitrile solution (50 ng ml$^{-1}$ tolbutamide in MeOH); a double blank sample was prepared with 200 µl 50% methanol acetonitrile solution. Samples were centrifuged at 1,790 × $g$ for 10 min at 4 °C. Then, 100 µl of the supernatant was transferred to a clean tube to which 100 µl water was added. The calibration standards were prepared by spiking different concentrations of RAY1216 or PF-07321332 with blank plasma. The concentrations of RAY1216 and PF-07321332 were determined using LC−MS/MS (Triple Quad 5500+).

RAY1216 samples were separated using a Synergi 4 µm Fusion-RP 80A LC column 50 × 2 mm at a temperature of 40 °C. The gradient mode consisted of mobile phases A (0.1% formic acid in water) and B (acetonitrile) as follows: 0.40 min (85% A and 15% B), 1.10 and 1.60 min (2% A and 98% B), and 1.61 and 2.20 min (85% A and 15% B). The flow rate was 0.7 ml min$^{-1}$. Prepared samples, 3 µl, were injected for analysis. PF-07321332 samples were separated using a Synergi 4 µm Fusion-RP 80A LC column 50 × 2 mm at a temperature of 40 °C. The gradient mode consisted of mobile phases A (0.1% formic acid in water) and B (acetonitrile) as follows: 0.20 min (80% A and 20% B), 1.60 and 1.90 min (5% A and 95% B), and 1.91 and 2.20 min (80% A and 20% B). The flow rate was 0.7 ml min$^{-1}$. Prepared samples, 3 µl, were injected for analysis.

## LC−MS/MS analysis of rat plasma samples

For the rat pharmacokinetic study, 50 µl RAY1216 plasma samples (the blank sample used 50 µl blank plasma) were mixed with 500 µl of 50% methanol acetonitrile solution (50 ng ml$^{-1}$ tolbutamide in MeOH); a double blank sample was prepared with 500 µl 50% methanol acetonitrile solution. Then, 40 µl PF-07321332 plasma samples (the blank sample used 40 µl blank plasma) were mixed with 400 µl of 50% methanol acetonitrile solution (50 ng ml$^{-1}$ tolbutamide in MeOH); a double blank sample was prepared with 400 µl 50% methanol acetonitrile solution. All the samples were centrifuged at 1,790 × $g$ for 10 min at 4 °C. Subsequently, 100 µl of the supernatant was transferred to a clean tube to which 100 µl water was added. The calibration standards were prepared by spiking different concentrations of RAY1216 or PF-07321332 with blank plasma. The concentrations of RAY1216 and PF-07321332 were determined using LC−MS/MS (Triple Quad 6500+).

RAY1216 samples were separated using a Waters XBridge BEH C4 2.5 µm column with a temperature of 40 °C. The gradient mode consisted of mobile phases A (0.1% formic acid in water) and B (acetonitrile) as follows: 0.20 min (92% A and 8% B), 0.60 min (45% A and 55% B), 1.60 min and 1.90 min (2% A and 98% B), and 1.91 and 2.20 min (92% A and 8% B). The flow rate was 0.8 ml min$^{-1}$. Prepared samples, 2 µl each, were injected for analysis. PF-07321332 samples were separated using a Synergi 4 µm Fusion-RP 80A LC column 50 × 2 mm with a temperature

of 40 °C. The gradient mode consisted of mobile phases A (0.1% formic acid in water) and B (acetonitrile) as follows: 1.60 min and 1.90 min (5% A and 95% B), and 1.91 min and 2.20 min (80% A and 20% B). The flow rate was 0.8 ml min$^{-1}$. Prepared samples, 1 µl, were injected for analysis.

## SARS-CoV-2 replicon inhibition assay

A previously described SARS-CoV-2 replicon system was used[51]. Briefly, the gene encoding wild-type or mutant M$^{pro}$ was cloned into the ps2AC vector expressing Nsp5 (M$^{pro}$). ps2V (0.1 µg), ps2AN (0.05 µg), ps2AC (0.4 µg) and ps2B (0.4 µg) were co-transfected into HEK293T cells (cell density: $6.5 × 10^4$ cm$^{-2}$) seeded in a 12-well plate. Then, 24 h after transfection, the cells were washed with PBS before the medium containing RAY1216 or PF-07321332 at different concentrations was added. Luciferase assay was performed 24 h after drug addition. Percentage inhibition was normalized to the luciferase activity of the DMSO control wells. Dose−response curves were plotted in GraphPad Prism 8.0 and were fit to a four-parameter variable-slope dose−response equation.

## Reporting summary

Further information on research design is available in the Nature Portfolio Reporting Summary linked to this article.

## Data availability

The single-crystal X-ray structure of RAY1216 has been deposited in The Cambridge Crystallographic Data Centre (www.ccdc.cam.ac.uk, CCDC) with the CDS Entry number DIDVEV and CCDC number 2251675 (ref. 64). The data can be obtained free of charge from CCDC via www.ccdc.cam.ac.uk/data_request/cif. The coordinates and structure factors of M$^{pro}$ crystal structures have been deposited in the Protein Data Bank (www.wwpdb.org) under accession numbers 8IGO (Apo M$^{pro}$) and 8IGN (RAY1216:M$^{pro}$). The raw LC−MS/MS data for each plasma sample from the SD rat, ICR mouse and K18-ACE2 mouse pharmacokinetic studies and the RAY1216 plasma stability data are available as Supplementary data. Source data are provided with this paper.

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

## Acknowledgements

We thank the staff at beamline BL19U1 of the Shanghai Synchrotron Radiation Facility for assistance on data collection. We thank L. Hedstrom, Brandeis University, for helpful comments on an early version of the paper. This work was supported by the National Multidisciplinary Innovation Team Project of Traditional Chinese Medicine (ZYYCXTU-D-202206 to Z.Y.), National Natural Science Foundation of China (82341085 to X.X.; 82341099 to Z.Y.), Guangdong Science and Technology Foundation (2022B1111060003 to Z.Y.; 2022A1515010301 and 2020A1515111155 to Q.M.), Guangzhou Science and Technology Planning Project (2022B01W0001 and 202201020449 to Q.M.; 202201020523 to Z.Y.), Emergency Key Programme of Guangzhou Laboratory (EKPG21-06 to Z.Y.), R&D Program of Guangzhou Laboratory (SRPG22-002 and SRPG22-003 to X.X.; TL22-13 to Z.Y.), The Young Top Talent of Science and Technology Innovation Department of Guangdong Province (2021TQ060189 to Q.M.), Macao Science and Technology Development Fund (0022/2021/A1 to N.Z.), The Youth Lift Project of China Association for Science and Technology (2020-2022QNRC001 to Q.M.), Natural Science Fund of Guangdong Province (2021A1515011289 to X.X.) and National Key Research and Development Program of China (2021YFA1300903 to X.X.; 2021YFA1300904 to T.Z.; 2022YFC0868700 to Xiaoxin Chen). X.X. acknowledges start-up grants from the Chinese Academy of Sciences. The funders had no role in study design, data collection and analysis, decision to publish or preparation of the paper.

## Author contributions

Z.Y., X.X. and Xiaoxin Chen conceived the study under the direction of N.Z. Xiaoxin Chen, J.H., H.L., C.L., Z.Z. and S.-H.C. provided the M^pro inhibitors and collected chemical characterization data and in vivo and in vitro pharmacokinetic data. X.H. expressed and purified M^pro, and performed enzyme kinetic assays under the supervision of X.X. Q.M. performed virus inhibition assays in cell culture and animal models and prepared figures with assistance from B.L., H.J., W.Z., C.Y. and S.W. and under the supervision of Z.Y. S.Z. and J.C. performed replicon inhibition studies and prepared figures under the supervision of Xinwen Chen. P.K., X.H. and X.X. analysed enzyme kinetic data and prepared figures. X.H. and B.Z. obtained M^pro crystals and performed crystal soaking experiments under the supervision of X.X. J.X., X.H., B.Z., Y.S. and Y.G. performed M^pro crystal diffraction experiments and collected diffraction data under the supervision of Y.X. and J.L. X.H. determined the M^pro crystal structures and built molecular models with assistance from C.N. and L.X. and under the supervision of X.X. X.H. and X.X. analysed the M^pro crystal structures and prepared figures, with input from all authors. X.X., P.K., X.H., Q.M. and Xiaoxin Chen wrote the initial draft, which was reviewed and edited by Z.Y., S.-H.C., Z.X., J.S., T.Z., J.H. and W.D. N.Z., Xinwen Chen, X.X. and Z.Y. acquired funding and supervised the research.

## Competing interests

Xiaoxin Chen, J.H., H.L. and C.L. are employees of Guangdong Raynovent Biotech, which holds the patent of RAY1216. Guangdong Raynovent Biotech provided the RAY1216 molecule used in this study and was responsible for the chemical characterization of RAY1216 and the in vivo and in vitro pharmacokinetic studies. The Raynovent company is not involved in the interpretation of the other results reported in this study and provides no funding to the other parties. P.K. is the author and distributor of the software package DynaFit. DynaFit licenses are available free of charge to all academic, educational and non-profit research institutions. The other authors declare no competing interests.

## Additional information

**Extended data** is available for this paper at https://doi.org/10.1038/s41564-024-01618-9.

**Correspondence and requests for materials** should be addressed to Shu-Hui Chen, Nanshan Zhong, Xiaoli Xiong or Zifeng Yang.

[1]School of Pharmaceutical Sciences (Shenzhen), Sun Yat-sen University, Shenzhen, China. [2]Guangdong Raynovent Biotech Co., Ltd, Guangzhou, China. [3]State Key Laboratory of Respiratory Disease, National Clinical Research Center for Respiratory Disease, Guangzhou Institute of Respiratory Health, The First Affiliated Hospital of Guangzhou Medical University, Guangzhou, China. [4]State Key Laboratory of Respiratory Disease, Guangdong Provincial Key Laboratory of Stem Cell and Regenerative Medicine, Guangdong Provincial Key Laboratory of Biocomputing, Guangdong Provincial Key Laboratory of Stem Cell and Regenerative Medicine, GIBH-CUHK Joint Research Laboratory on Stem Cell and Regenerative Medicine; Guangzhou Institutes of Biomedicine and Health, Chinese Academy of Sciences, Guangzhou, China. [5]BioKin Ltd, Watertown, MA, USA. [6]Guangzhou National Laboratory, Guangzhou, China. [7]Guangdong Provincial Key Laboratory of Chemical Measurement and Emergency Test Technology, Institute of Analysis, Guangdong Academy of Sciences (China National Analytical Center Guangzhou), Guangzhou, China. [8]School of Life Sciences, University of Science and Technology of China, Hefei, China. [9]School of Cellular and Molecular Medicine, University of Bristol, Bristol, UK. [10]WuXi AppTec, Shanghai, China. [11]State Key Laboratory of Virology, Wuhan Institute of Virology, Center for Biosafety Mega-Science, Chinese Academy of Sciences, Wuhan, China. [12]State Key Laboratory of Quality Research in Chinese Medicine, Macau Institute for Applied Research in Medicine and Health, Macau University of Science and Technology, Macau (SAR), China. [13]These authors contributed equally: Xiaoxin Chen, Xiaodong Huang, Qinhai Ma, Petr Kuzmič. ✉e-mail: chen_shuhui@wuxiapptec.com; nanshan@vip.163.com; xiong_xiaoli@gibh.ac.cn; jeffyah@163.com

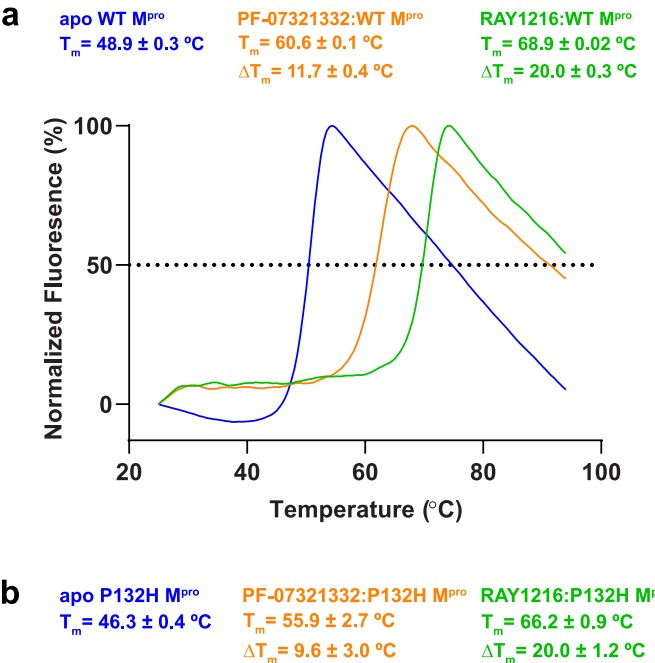

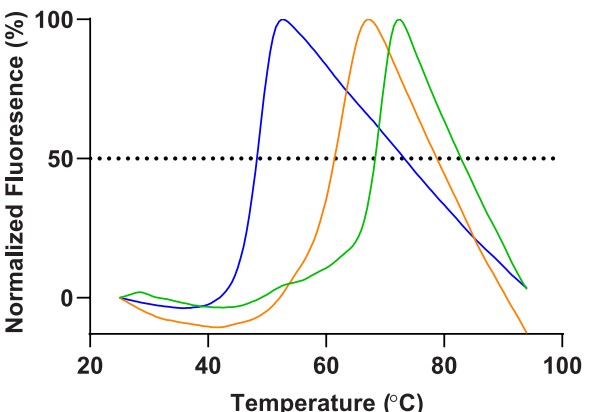

**Extended Data Fig. 1 | Thermal melting curves of WT and P132H M^pro proteins and their inhibitor complexes.** Experiments were carried out at an M^pro concentration of 3 μM. For the inhibitor complexes, 30 min incubation at 16 °C with 6 μM RAY1216 (green) or PF-07321332 (orange) was performed before the melting curves were recorded. **a**, WT M^pro proteins and their complexes. **b**, P132H M^pro proteins and their inhibitor complexes. $T_m$ values and thermal shift $\Delta T_m$ values are shown as mean ± SD ($n = 3$).

## a Antiviral activity of RAY1216 by CPE inhibition assay

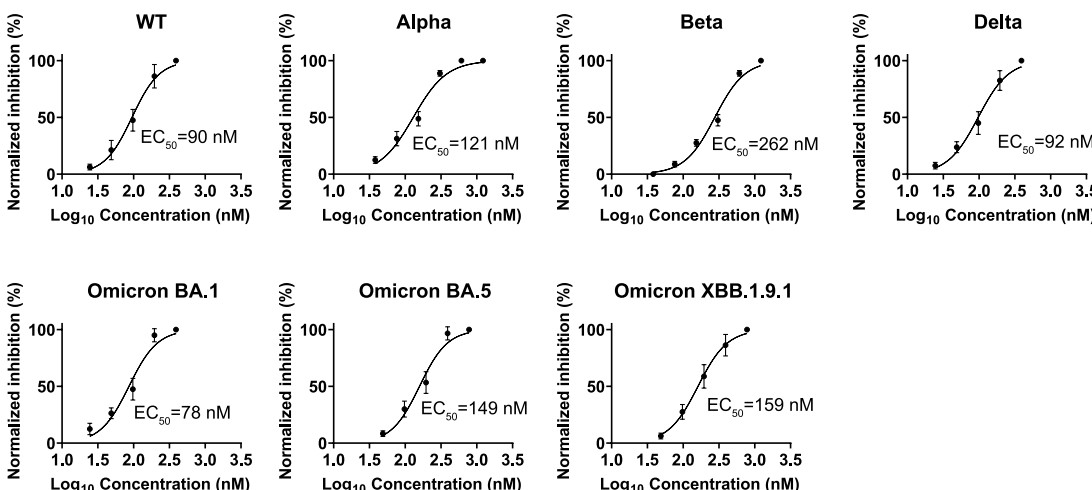

## b Antiviral activity of PF-07321332 by plaque reduction assay

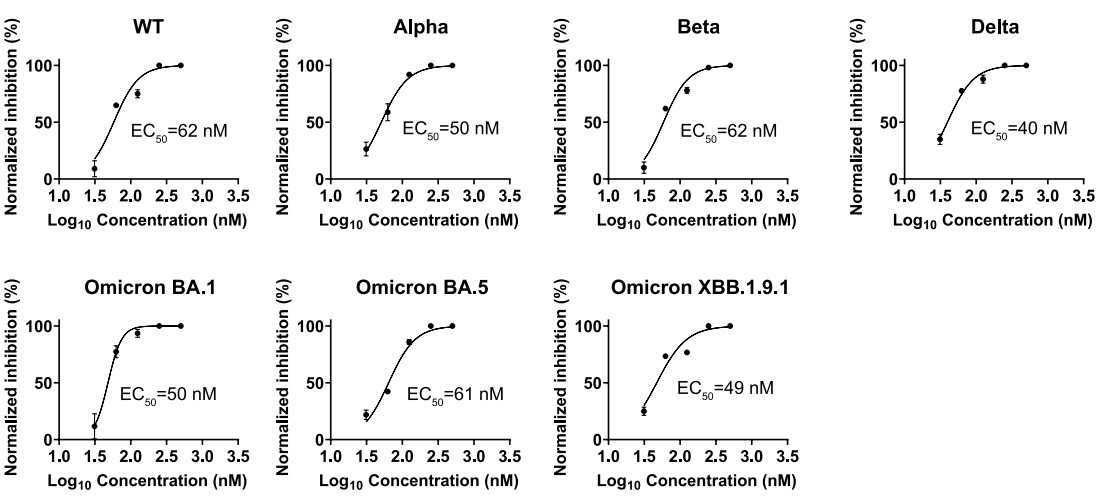

## c Antiviral activity of PF-07321332 by CPE inhibition assay

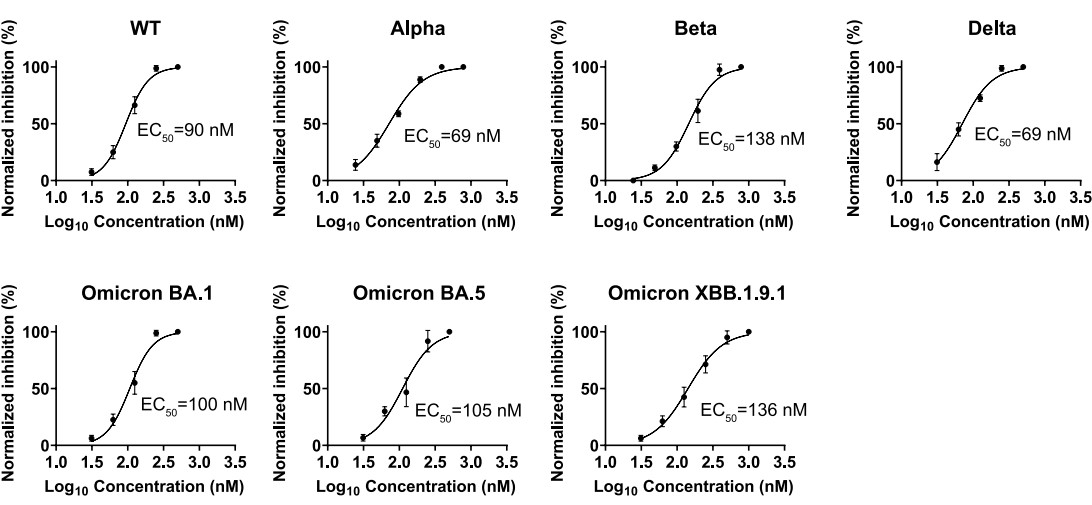

**Extended Data Fig. 2 | Virus inhibition by RAY1216 and PF-07321332. a**, Virus inhibition dose–response curves of RAY1216 as assessed by cytopathic effect (CPE) inhibition assay. **b** and **c**, virus inhibition dose–response curves of PF-07321332 as assessed by plaque reduction or CPE inhibition assay. All data points are shown as mean ± SD, $n = 3$.

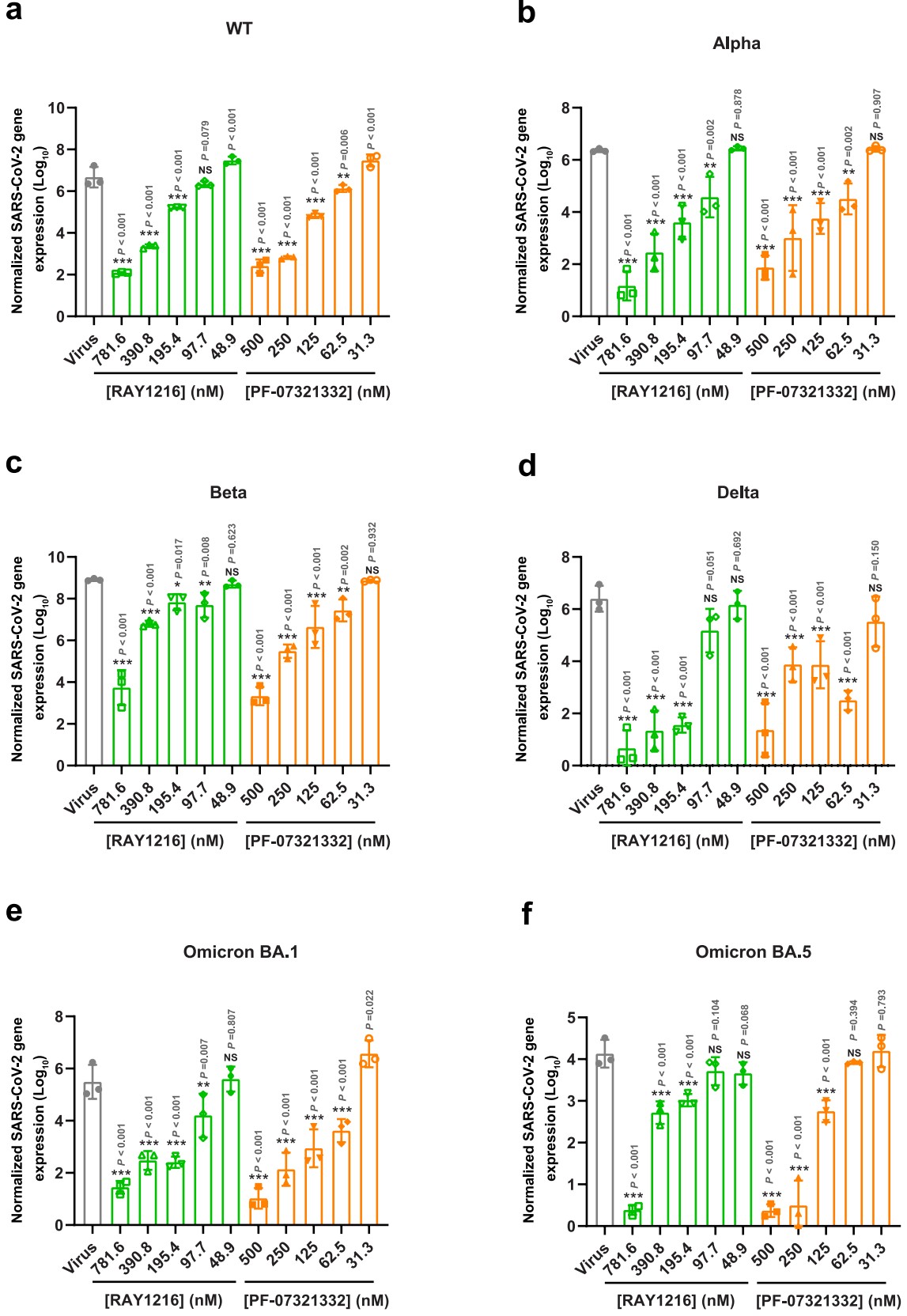

**Extended Data Fig. 3 | Inhibition of SARS-CoV-2 variants by RAY1216 and PF-07321332 as assessed by qPCR.** Viral gene in the supernatants of Vero E6 cell infected by wildtype (**a**), Alpha (**b**), Beta (**c**), Delta (**d**), Omicron BA.1 (**e**) and Omicron BA.5 (**f**) SARS-CoV-2 authentic viruses in the presence of different concentrations of RAY1216/PF-07321332 was measured by quantitative real-time PCR assay (mean±SD, $n = 3$). *$P \leq 0.05$; **$P \leq 0.01$; ***$P \leq 0.001$; NS, not significant as determined by one-way ANOVA analysis of variance with Tukey's honest significant difference (HSD) test compared with the virus group.

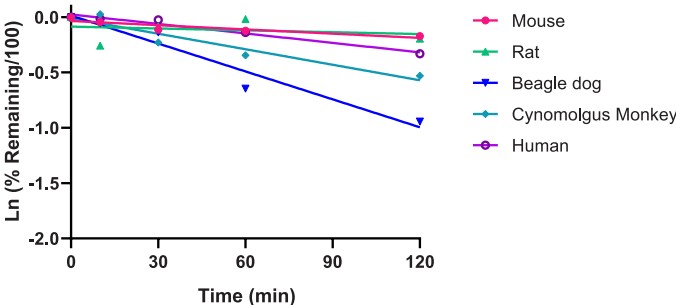

**Extended Data Fig. 4 | Stability of RAY1216 in plasmas of different species.** Data points showing natural log of fraction remaining are plotted as a function of time. Lines show linear regression of the data points for each plasma.

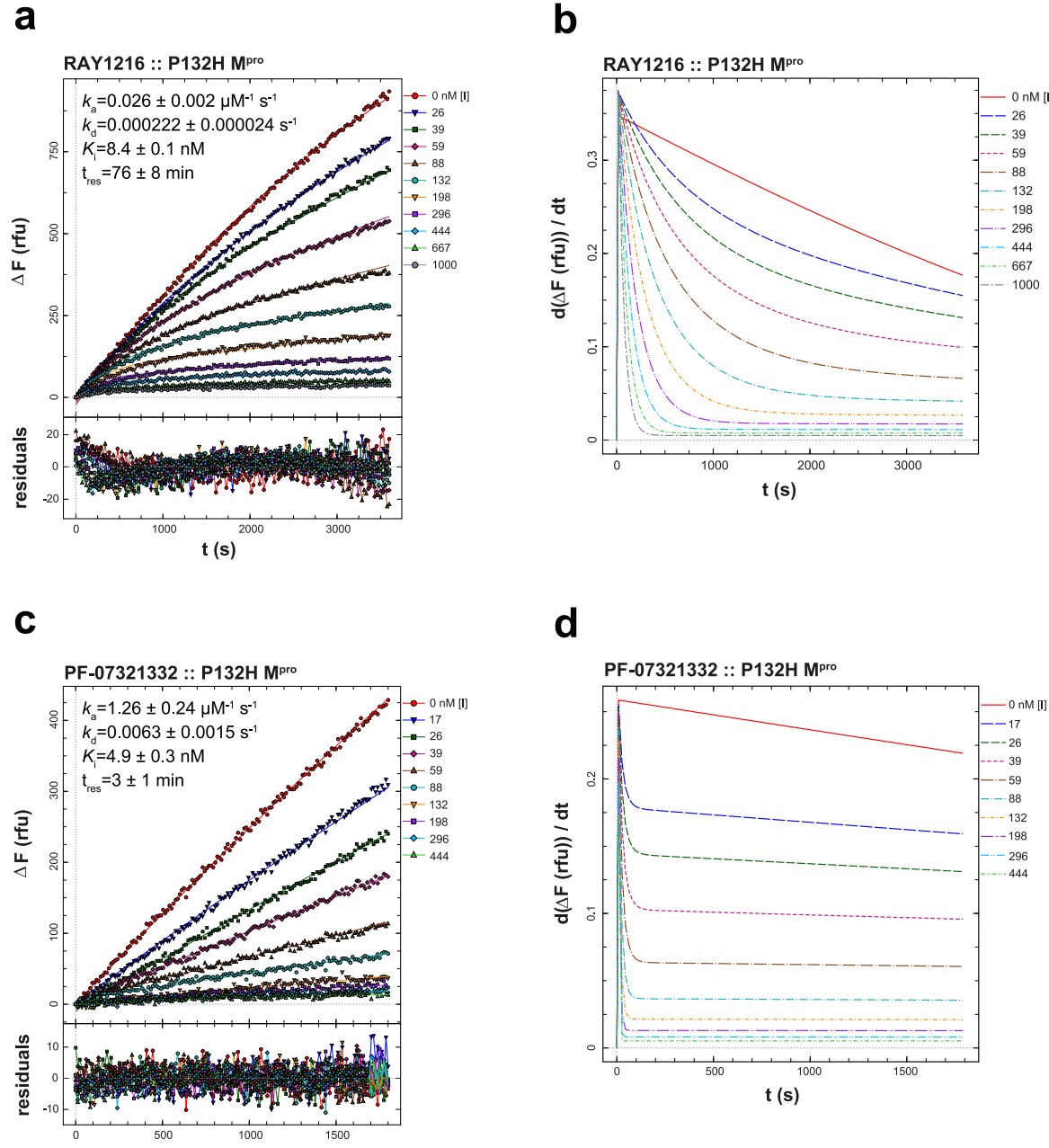

**Extended Data Fig. 5 | Inhibition of P132H M$^{pro}$ by RAY1216 and PF-07321332.** **a**, RAY1216 inhibition progress curves (dots) are overlaid with best-fit model curves; the residuals of the fits are shown. **b**, Plots of instantaneous reaction rates for the progress curves in panel a. **c**, PF-07321332 inhibition progress curves (dots) are overlaid with best-fit model curves; the residuals of the fits are shown. **d**, Plots of instantaneous reaction rates for the progress curves in panel **c**. See *Enzyme kinetics section of Supplementary Information* for details of data analysis procedures.

**Extended Data Table 1 | RAY1216 protease specificity**

| Protease | RAY1216 IC$_{50}$ (µM) | Reference inhibitor | Reference inhibitor IC$_{50}$ (µM) |
|---|---|---|---|
| **SARS-CoV-2 M$^{pro}$** | 0.033 | - | - |
| **hCathepsin B** | 4.6 | **Leupeptin hemisulfate** | 0.059 |
| **hChymotrypsin C** | >100 | **Chymostatin** | 0.234 |
| **hCathepsin D** | >100 | **Pepstatin A** | 0.18 |
| **hElastase** | 84.9 | **BAY-678** | 0.022 |

RAY1216 IC$_{50}$ values against human cathepsin B, chymotrypsin C, cathepsin D and elastase are shown and compared to those of reference inhibitors.

**Extended Data Table 2 | Kinetic parameters (mean and standard deviation from replicates, *n* = 3) of M^pro inhibition by RAY1216 and PF-07321332 as determined by ODE method in DynaFit**

### a

## Kinetic parameters of WT M^pro inhibition

| parameter | unit | RAY1216 | PF-07321332 | ratio RAY/PF |
|---|---|:---:|:---:|:---:|
| $k_a$ | µM$^{-1}$ s$^{-1}$ | 0.019 ± 0.001 | 0.49 ± 0.04 | 0.04 |
| $k_d$ | s$^{-1}$ | 0.000161 ± 0.000008 | 0.0018 ± 0.0002 | 0.09 |
| $K_i = k_d/k_a$ | nM | 8.4 ± 0.2 | 3.8 ± 0.2 | 2.2 |
| $t_{res}$ | min | 104 ± 5 | 9 ± 1 | 11.5 |

### b

## Kinetic parameters of P132H M^pro inhibition

| parameter | unit | RAY1216 | PF-07321332 | ratio RAY/PF |
|---|---|:---:|:---:|:---:|
| $k_a$ | µM$^{-1}$ s$^{-1}$ | 0.026 ± 0.002 | 1.26 ± 0.24 | 0.02 |
| $k_d$ | s$^{-1}$ | 0.000222 ± 0.000024 | 0.0063 ± 0.0015 | 0.04 |
| $K_i = k_d/k_a$ | nM | 8.4 ± 0.1 | 4.9 ± 0.3 | 1.7 |
| $t_{res}$ | min | 76 ± 8 | 3 ± 1 | 25.3 |

a, Kinetic parameters of WT M^pro inhibition. b, Kinetic parameters of P132H M^pro inhibition. See *Enzyme kinetics section of Supplementary Information* for details of data analysis procedures.

**Extended Data Table 3 | Stability of RAY1216 and formation of the P1 R-epimer 'RAY1216-E' in plasmas of different species**

| | Time point (min) | Mouse | Rat | Beagle dog | Cynomolgus Monkey | Human |
|---|---|---|---|---|---|---|
| **%Remaining of RAY1216** | **0** | 100.0 | 100.0 | 100.0 | 100.0 | 100.0 |
| | **10** | 95.5 | 78.1 | 95.0 | 103.3 | 98.7 |
| | **30** | 89.3 | 92.1 | 88.8 | 79.9 | 97.5 |
| | **60** | 88.5 | 98.2 | 52.4 | 71.4 | 88.4 |
| | **120** | 84.2 | 82.3 | 38.9 | 58.8 | 71.8 |
| | **$T_{1/2}$ (min)** | **>289.1** | **>289.1** | **82.1** | **147.8** | **244.9** |
| | Time point (min) | Mouse | Rat | Beagle dog | Cynomolgus Monkey | Human |
| **%Formation of RAY1216-E** | **0** | 1.4 | 1.4 | 0.4 | 0.7 | 1.4 |
| | **10** | 2.7 | 1.3 | 4.8 | 14.9 | 5.1 |
| | **30** | 3.0 | 1.8 | 22.5 | 19.9 | 9.5 |
| | **60** | 3.8 | 2.7 | 35.1 | 19.1 | 13.4 |
| | **120** | 4.0 | 2.6 | 51.9 | 13.2 | 10.6 |

Percentage (%) remaining of RAY1216 and percentage (%) formation of RAY1216-E at different time points in plasmas of different species are reported. Half-lives ($T_{1/2}$) of RAY1216 in plasmas of different species are reported in minutes (min).

**Extended Data Table 4 | Pharmacokinetic parameters of RAY1216 and PF-07321332 after intravenous injection (i.v.) dosing and gavage (p.o.) dosing in mouse and rat models**

| Compound | Species | dose (mg/kg) | Gender | $C_{max}$ (ng/mL) | $T_{max}$ (h) | $AUC_{0\sim last}$ (h·ng/mL) | Cl (mL/min/kg) | $Vd_{ss}$ (L/kg) | $T_{1/2}$ (h) | oral F (%) |
|---|---|---|---|---|---|---|---|---|---|---|
| RAY1216 | Mouse | 3.0 (i.v.) | Male | 4400 ± 270 | 0.083 ± 0.008 | 5300 ± 860 | 9.6 ± 1.7 | 1.0 ± 0.1 | 3.3 ± 1.2 | -- |
| | | | Female | 4900 ± 580 | 0.083 ± 0.008 | 7000 ± 760 | 7.2 ± 0.8 | 0.8 ± 0.1 | 3.7 ± 0.5 | -- |
| | | 10 (p.o.) | Male | 2200 ± 290 | 0.4 ± 0.1 | 5600 ± 1400 | -- | -- | 1.8 ± 0.6 | 32 ± 8 |
| | | | Female | 3300 ± 300 | 0.5 ± 0.3 | 9200 ± 1400 | -- | -- | 2.7 ± 0.2 | 39 ± 6 |
| | Rat | 2.0 (i.v.) | Male | 3200 ± 520 | 0.083 ± 0.008 | 4500 ± 680 | 7.4 ± 0.9 | 0.8 ± 0.1 | 4.8 ± 0.6 | -- |
| | | | Female | 2400 ± 300 | 0.083 ± 0.008 | 3400 ± 600 | 10.0 ± 2.0 | 1.1 ± 0.2 | 4.2 ± 0.5 | -- |
| | | 10 (p.o.) | Male | 1100 ± 260 | 1.1 ± 1.0 | 6300 ± 1100 | -- | -- | 3.5 ± 0.4 | 28 ± 5 |
| | | | Female | 1000 ± 330 | 1.8 ± 1.5 | 5300 ± 890 | -- | -- | 3.0 ± 0.3 | 31 ± 5 |
| PF-07321332 | Mouse | 3.0 (i.v.) | Male | 2300 ± 280 | 0.083 ± 0.008 | 1000 ± 130 | 50.0 ± 6.8 | 0.9 ± 0.1 | 0.25 ± 0.04 | -- |
| | | | Female | 2300 ± 150 | 0.083 ± 0.008 | 940 ± 160 | 53.7 ± 7.7 | 0.9 ± 0.2 | 0.2 ± 0.1 | -- |
| | | 10 (p.o.) | Male | 890 ± 260 | 0.12 ± 0.07 | 810 ± 170 | -- | -- | 0.7 ± 0.3 | 24 ± 5 |
| | | | Female | 1300 ± 380 | 0.12 ± 0.07 | 1200 ± 420 | -- | -- | 0.7 ± 0.3 | 39 ± 13 |
| | Rat | 2.0 (i.v.) | Male | 1700 ± 220 | 0.083 ± 0.008 | 1100 ± 180 | 46.5 ± 8.0 | 1.5 ± 0.1 | 0.5 ± 0.1 | -- |
| | | | Female | 2300 ± 380 | 0.083 ± 0.008 | 1500 ± 280 | 34.6 ± 6.2 | 1.5 ± 0.3 | 1.2 ± 0.5 | -- |
| | | 10 (p.o.) | Male | 630 ± 180 | 0.9 ± 0.7 | 1600 ± 600 | -- | -- | 1.1 ± 0.4 | 30 ± 11 |
| | | | Female | 940 ± 160 | 0.4 ± 0.1 | 2200 ± 860 | -- | -- | 1.0 ± 0.3 | 30 ± 12 |

**Notes:** $C_{max}$: The maximum observed concentration of the drug collected in bodily material from the tested animal

$T_{max}$: The time it takes to reach the maximum concentration $C_{max}$

**AUC**: "Area Under the Curve" represents the total exposure of the drug experienced by the tested animal

**Cl**: Total plasma clearance

$Vd_{ss}$: Steady state volume of distribution

$T_{1/2}$: Half-life, which is the time it takes for the drug concentration to decrease by a half

**oral (F%)**: Oral bioavailability

Data are shown as mean ± SD (*n* = 5).

**Extended Data Table 5 | Inhibition titres (EC$_{50}$) of RAY1216 and PF-07321332 in SARS-CoV-2 replicon assays**

| M$^{pro}$ variant | EC$_{50}$ (nM) | |
| --- | --- | --- |
| | **RAY1216** | **PF-07321332** |
| **WT** | 50 ± 6 | 33 ± 2 |
| **G15S** | 136 ± 11 (2.7x) | 78 ± 10 (2.4x) |
| **M49L** | 258 ± 18 (5.2x) | 163 ± 8 (4.9x) |
| **F140L** | 424 ± 15 (8.5x) | 295 ± 9 (8.9x) |
| **△P168** | 568 ± 18 (11.4x) | 422 ± 19 (12.8x) |
| **L50F/E166V** | 233 ± 4 (4.7x) | >5000 (>150x) |
| **E166A/L167F** | 327 ± 21 (6.5x) | 540 ± 38 (16.4x) |

The EC$_{50}$ values are shown as mean ± SD ($n$ = 3) for three biologically independent SARS-CoV-2 replicon assay experiments; fold changes relative to EC$_{50}$ values determined against the replicon encoding WT M$^{pro}$ are shown in the parentheses.

# Reporting Summary

## Statistics

For all statistical analyses, confirm that the following items are present in the figure legend, table legend, main text, or Methods section.

| n/a | Confirmed | |
|---|---|---|
| ☐ | ☒ | The exact sample size (*n*) for each experimental group/condition, given as a discrete number and unit of measurement |
| ☐ | ☒ | A statement on whether measurements were taken from distinct samples or whether the same sample was measured repeatedly |
| ☐ | ☒ | The statistical test(s) used AND whether they are one- or two-sided<br>*Only common tests should be described solely by name; describe more complex techniques in the Methods section.* |
| ☒ | ☐ | A description of all covariates tested |
| ☒ | ☐ | A description of any assumptions or corrections, such as tests of normality and adjustment for multiple comparisons |
| ☐ | ☒ | A full description of the statistical parameters including central tendency (e.g. means) or other basic estimates (e.g. regression coefficient) AND variation (e.g. standard deviation) or associated estimates of uncertainty (e.g. confidence intervals) |
| ☐ | ☒ | For null hypothesis testing, the test statistic (e.g. *F*, *t*, *r*) with confidence intervals, effect sizes, degrees of freedom and *P* value noted<br>*Give P values as exact values whenever suitable.* |
| ☒ | ☐ | For Bayesian analysis, information on the choice of priors and Markov chain Monte Carlo settings |
| ☒ | ☐ | For hierarchical and complex designs, identification of the appropriate level for tests and full reporting of outcomes |
| ☒ | ☐ | Estimates of effect sizes (e.g. Cohen's *d*, Pearson's *r*), indicating how they were calculated |

*Our web collection on statistics for biologists contains articles on many of the points above.*

## Software and code

Policy information about availability of computer code

| | |
|---|---|
| Data collection | Molecular device SoftMax Pro 7.0 Software was used to measure enzyme kinetics data and luminescence in the replicon assay. ABI PRISM 7500 real-time PCR System (Applied Biosystems) was used to measure RNA levels in the virus inhibition assay. |
| Data analysis | DynaFit 4, XDS software (BUILT 20220220), CCP4 7.1.018, Phaser MR 2.8.3, Coot 0.9.6, Refmac 5.8.0267, Phoenix WinNonlin software (version 8.2.0), SHELXT, SHELXL. IBM SPSS Statistics Version 25.0. GraphPad Prism version 8.0 |

For manuscripts utilizing custom algorithms or software that are central to the research but not yet described in published literature, software must be made available to editors and reviewers. We strongly encourage code deposition in a community repository (e.g. GitHub). See the Nature Portfolio guidelines for submitting code & software for further information.

## Data

Policy information about availability of data

All manuscripts must include a data availability statement. This statement should provide the following information, where applicable:
- Accession codes, unique identifiers, or web links for publicly available datasets
- A description of any restrictions on data availability
- For clinical datasets or third party data, please ensure that the statement adheres to our policy

The single crystal X-ray structure of RAY1216 has been deposited in The Cambridge Crystallographic Data Centre (www.ccdc.cam.ac.uk ,CCDC) with the CDS Entry number DIDVEV and CCDC number 2251675 (DOI: 10.5517/ccdc.csd.cc2fl1ps). The data can be obtained free of charge from CCDC via www.ccdc.cam.ac.uk/

data_request/cif. The coordinates and structure factors of Mpro crystal structures have been deposited in the Protein Data Bank (www.wwpdb.org) under accession numbers 8IGO (Apo Mpro) and 8IGN (RAY1216:Mpro). Source data are provided with this paper.

## Human research participants

Policy information about studies involving human research participants and Sex and Gender in Research.

| | |
|---|---|
| Reporting on sex and gender | Not applicable. |
| Population characteristics | Not applicable. |
| Recruitment | Not applicable. |
| Ethics oversight | Not applicable. |

Note that full information on the approval of the study protocol must also be provided in the manuscript.

# Field-specific reporting

Please select the one below that is the best fit for your research. If you are not sure, read the appropriate sections before making your selection.

☒ Life sciences  ☐ Behavioural & social sciences  ☐ Ecological, evolutionary & environmental sciences

For a reference copy of the document with all sections, see nature.com/documents/nr-reporting-summary-flat.pdf

# Life sciences study design

All studies must disclose on these points even when the disclosure is negative.

| | |
|---|---|
| Sample size | The sample sizes were similar to those reported in previous publications (https://doi.org/10.1038/s41586-020-2312-y and https://doi.org/10.1126/science.abf1611 ).For mouse model using in the antiviral study, there were seven mice in each group (six groups in total). For the ICR mouse model in vivo pharmacokinetic study, there were five male mice and five female mice per group in different sets of experiments. For the SD rat model in vivo pharmacokinetic study, there were five male rats and five female rats per group in different sets of experiments. For the K18-hACE2 mouse model in vivo pharmacokinetic study,  there were five K18-hACE2 female mice per group in different sets of experiments. This is to meet the requirement for statistical analysis while ensuring good technical reproducibility. Other assays were performed for duplicates or three replicates, which were also sufficient for a good statistical analysis. |
| Data exclusions | No data were excluded from the analyses presented in the manuscript. |
| Replication | As for the plasma concentration, viral titers, and HE stain experiments, at least 3 animals for each group (at each detecting time point) were tested, and some of them were tested in two or four replications. All attempts at replication were successful. The replicates of the enzyme kinetic assay and the experiments based on the cells were noted in the figure legends and methods. |
| Randomization | For the antiviral study, we randomly divided 42 K18-hACE2 transgenic mice into six groups. For lung tissues analyzed with histological staining and virus titer determination assays, we chose a specific number of mice for examinations, and histological images were selected randomly from the corresponding experimental groups. For the pharmacokinetic study in vivo, there were no randomizations in dividing ICR mice, SD rats, and K18-hACE2 mice into different groups. |
| Blinding |  The investigators were not blinded to allocation during experiments andoutcome assessment. The relevant quantitative experiments in the manuscript, such as the determination of virus titer, viral gene level expression, drug concentrations in plasma, and the records the changes of the animals' weight and survival rate, need to be correctly and clearly labeled on the tubes or cages. All samples are tested and analyzed in accordance with the protocol, and the results would not be effected by the subjective judgment of the investigators. The evaluation of histopathological changes required a qualified and experienced pathologist to observe all samples. The results of the comprehensive evaluation of all samples in different groups were described in the manuscript to exclude personal subjective bias. |

# Reporting for specific materials, systems and methods

We require information from authors about some types of materials, experimental systems and methods used in many studies. Here, indicate whether each material, system or method listed is relevant to your study. If you are not sure if a list item applies to your research, read the appropriate section before selecting a response.

## Materials & experimental systems

| n/a | Involved in the study |
|-----|----------------------|
| ☒ | Antibodies |
| ☐ | ☒ Eukaryotic cell lines |
| ☒ | Palaeontology and archaeology |
| ☐ | ☒ Animals and other organisms |
| ☒ | Clinical data |
| ☒ | Dual use research of concern |

## Methods

| n/a | Involved in the study |
|-----|----------------------|
| ☒ | ChIP-seq |
| ☒ | Flow cytometry |
| ☒ | MRI-based neuroimaging |

# Eukaryotic cell lines

Policy information about cell lines and Sex and Gender in Research

| | |
|---|---|
| Cell line source(s) | Vero E6 cells: ATCC, CRL-1586; HEK 293T cells: ATCC,CRL-3216. |
| Authentication | All cell lines were frequently checked for the cellular morphologies, growth rates and functions in our lab and were not commonly misidentified. |
| Mycoplasma contamination | The cell lines were not contaminated by mycoplasma as determined by using the Lonza Mycoplasma Detection Kit. |
| Commonly misidentified lines (See ICLAC register) | No commonly misidentified cell lines were used. |

# Animals and other research organisms

Policy information about studies involving animals; ARRIVE guidelines recommended for reporting animal research, and Sex and Gender in Research

| | |
|---|---|
| Laboratory animals | For the SARS-CoV-2 animal experiment, the five to six-weeks old female K18-hACE2-transgenic C57BL/6 mice were provided by Gempharmatech Co., Ltd. (Jiangsu, China). The animals were fed every day with the fodder purchased from Beijing Keao Xieli Feed Co., Ltd. and the general quality standards, hygienic standards and conventional nutritional ingredient index requirements in feeds are tested in accordance with GB14924.2-2001 and GB14924.3-2010 standards. All work with live SARS-CoV-2 was conducted in the Biosafety Level 3 (BLS3) Laboratories. The mice were randomly divided into six groups (7 mice per group). All mice were kept in SPF (specific pathogen free) facilities.<br>For the pharmacokinetic in vivo study, the four to six-weeks old ICR mice and the six to eight-weeks old SD rats were provided by Vital River Laboratory Animal Technology Co., Ltd. (Beijing, China). The six-weeks old K18-hACE2-transgenic C57BL/6 mice were provided by Gempharmatech Co., Ltd. (Jiangsu, China). The animals were fed every day with the fodder purchased from Wuhan WQJX Bio-Technology Co., Ltd. All the animals were house in controlled temperature (20-26°C), humidity (40-70%) and lighting conditions (12 h light/ 12 h dark cycles). |
| Wild animals | The study did not involve wild animals. |
| Reporting on sex | All the K18-hACE2-transgenic C57BL/6 mice were female. All the ICR mice and SD rats for the pharmacokinetics study were half male and half female. |
| Field-collected samples | No sample is collected from the field. |
| Ethics oversight | The antiviral studies were approved by the Guangzhou Medical University Ethics Committee of Animal Experiments (IACUC certificate No.: GZL0008). The PK studies were approved by the ethics committee the Institute Animal Care and Use Committee (IACUC) of Precedo Pharmaceuticals Co., Ltd. The IACUC No. for rat PK studies and mouse (including ICR mouse and K18-hACE2 mouse) PK studies are IACUC-20230303-2 and IACUC-20230303-3, respectively. All plasmas used in the RAY1216 plasma stability experiment are from commercially sources. The plasmas of CD-1 mouse and SD rat were purchased from Vital River Laboratories (Beijing, China). The cynomolgus monkey plasma was purchased from Xishan Zhongke Laboratory Animal Co., Ltd (Suzhou, China) and the plasmas of beagle dog (#CAN00PLK2Y2N) and human (#HUMANPLK2P2N) were purchased from BioIVT (NY, USA). |

Note that full information on the approval of the study protocol must also be provided in the manuscript.

