## [Peer Review File · Nature Microbiology]

Peer Review Information

Journal: Nature Microbiology

Manuscript Title: Preclinical evaluation of the SARS-CoV-2 Mpro inhibitor RAY1216 with improved pharmacokinetics compared to nirmatrelvir

Corresponding author name(s): Professor Xiaoli Xiong

Reviewer Comments & Decisions:Decision Letter, initial version:

Message 28th April 2023

:

Dear Xiaoli,

Thank you for your patience while your manuscript "Inhibition mechanism and antiviral activity of an α -ketoamide based SARS-CoV-2 main protease inhibitor" was under peer-review at Nature Microbiology. It has now been seen by 3 referees, whose expertise and comments you will find at the end of this email. Although they find your work of some potential interest, they have raised a number of concerns that will need to be addressed before we can consider publication of the work in Nature Microbiology.

In particular, both reviewer #1 and #3 ask for more in vitro data to better characterize antiviral breadth in different cell lines and to characterize dosing of RAY1216. In addition, reviewer #3 asks to test development of resistance against RAY1216 using SARS-CoV-2 replicons and to compare antiviral activity of RAY1216 to PF-07321332 in vivo.

Should further experimental data allow you to address these criticisms, we would be happy to look at a revised manuscript.

Please include a data availability statement as a separate section after Methods but before references, under the heading "Data Availability". This section should inform readers about the availability of the data used to support the conclusions of your study. This information includes accession codes to public repositories (data banks for protein, DNA or RNA sequences, microarray, proteomics data etc...), references to source data published alongside the paper, unique identifiers such as URLs to data repository entries, or data set DOIs, and any other statement about data availability. At a minimum, you should include the following statement: "The data that support the findings of this study are available from the corresponding

3author upon request", mentioning any restrictions on availability. If DOIs are provided, we also strongly encourage including these in the Reference list (authors, title, publisher (repository name), identifier, year). For more guidance on how to write this section please see: <http://www.nature.com/authors/policies/data/data-availability-statements-data-citations.pdf>

* If you have not done so already we suggest that you begin to revise your manuscript so that it conforms to our Article format instructions at <http://www.nature.com/nmicrobiol/info/final-submission>. Refer also to any guidelines provided in this letter.

When submitting the revised version of your manuscript, please pay close attention to our [href="https://www.nature.com/nature-portfolio/editorial-policies/image-integrity">Digital Image Integrity Guidelines. and to the following points below:](https://www.nature.com/nature-portfolio/editorial-policies/image-integrity)

Note: This url links to your confidential homepage and associated information about manuscripts you may have submitted or be reviewing for us. If you wish to forward this e-mail to co-authors, please delete this link to your homepage first.

Nature Microbiology is committed to improving transparency in authorship. As part of our efforts in this direction, we are now requesting that all authors identified as 'corresponding author' on published papers create and link their Open Researcher

and Contributor Identifier (ORCID) with their account on the Manuscript Tracking System (MTS), prior to acceptance. This applies to primary research papers only. ORCID helps the scientific community achieve unambiguous attribution of all scholarly contributions. You can create and link your ORCID from the home page of the MTS by clicking on 'Modify my Springer Nature account'. For more information please visit www.springernature.com/orcid

If you wish to submit a suitably revised manuscript we would hope to receive it within 6 months. If you cannot send it within this time, please let us know. We will be happy to consider your revision, even if a similar study has been accepted for publication at Nature Microbiology or published elsewhere (up to a maximum of 6 months).

Reviewer Expertise:

Referee #1: Pharmacology
Referee #2: Mpro, Structure
Referee #3: Virology

Reviewer Comments:

Reviewer #1 (Remarks to the Author):

The manuscript from Chen, Huang and Kuzmic describes the pre-clinical characterization of RAY1216, a clinical candidate currently in phase 3 clinical trials for the treatment of COVID-19. Given that this is the first disclosure of such an advanced clinical molecule, this represents a potential publication of significance. The paper is extremely well written and referenced, drawing from, and reflecting a field that has expanded enormously over the last 18 months. The enzymology section is of particular note and excellence. The slow on/off characteristics for RAY1216 represents an interesting orthogonality that actually merited greater discussion of potential advantages or disadvantages to the eventual clinical profile of the molecule than was offered. Throughout the manuscript characterizes RAY1216 alongside PF-7321332, the prototype Mpro inhibitor and the anti-viral component of Paxlovid, the first authorized oral Mpro inhibitor for COVID-19.

I would ask the editor to consider the following in correspondence with the authors before their final decision on whether to publish. While the paper contains all the components one would expect of a manuscript, a number of clarifications/responses are needed before I would support publication.

Proof reading/clarifications:

P2 L96 either 'in a phase 3 clinical trial' or 'in phase 3 clinical trials'

P6 177 This is a beautiful section on the enzymology. Having differentiated RAY1216 from PF-7321332 at the on/off rate level, how might this help of hinder the in vivo or clinical performance of RAY1216 versus Paxlovid? The manuscript would benefit from such a discussion.

P8 L244 The anti-viral work is conducted in VeroE6 cells. In the Science paper by Owen et al on the characterization of PF-7321332, it is stated that this assay must be conducted in the presence of an efflux inhibitor (EI) CP-100356. There is no mention of it in the experimental procedure or figure legends. PF-7321332 is a significant Pgp substrate and anti-viral activity cannot be seen with the presence of the EI when using the VeroE6 system. 1. Did the authors use the EI? 2. If not, how is the control PF-7321332 data explained? 3. Is RAY1216 a Pgp substrate (would seem likely)?

P9 L275 Is plasmata a word? I looked up if was some type of plural of plasma. Did not appear to be, but I may stand corrected!

P8/9 In vivo model. It looks the authors decided to compare the two molecules at the same 600mg/kg QD dose. With different PK, potentially PD (due to on/off rate differences) and to a lesser degree potency, this is not the ideal way to compare efficacy for the two molecules. These were not the doses/dosing regime used for PF-7321332 in the comparative paper cited. As a result, RAY1216 looks less effective in viral load lowering, albeit not to significance in this study from Chen. PF-7321332 is capable of offering up to a 2.5 log unit drop in this model when dosed 1000mg/kg BID.

What is the maximum efficacy for RAY1216 at doses above 600mg/kg? Did toxicity/formulation preclude higher dosing? The authors should provide a mouse free drug exposure plot for both molecules over 24 hours at 600mg/kg QD with a reference concentration line of the VeroE6 EC90 for both molecules. How much time does each molecule spend above EC90 as free drug and if so, at what are the multiples over EC90 at 12 and 24 hours? This assumes that 600mg/kg PK is available for each molecule (Ideally satellite non-infected animals given the same dosing solution as the infected ones). If not, it should really be run. The 10mg oral mouse PK stated in the paper, rightly referencing multiples of EC90, is useful but the PK component of the doses used in the efficacy are critical to interpret the in vivo outcome. Do the slow/on off characteristics have any connection to the in vivo efficacy seen? In the bioanalysis did the authors see any epimerization of the molecule at the P1 center? Boceprevir is a mixture of diastereomers at P1 given the stereochemical instability of an α -ketoamide in that case. Was RAY1216 stable to epimerization at P1? It seems to have been synthesized as a single stereoisomer in the experimental section, but what is the stability in blood or overtime? This will affect the interpretation of the in vivo study as it could be a non-metabolic source of active compound decline. It is surprising that there is no mention of this potential property for α -ketoamides in the manuscript.

P9 L287 This was the only sentence that was a little sweeping or unsupported in the entire manuscript. It is difficult to credit α -ketoamides as having superior chemical and metabolic stability (implying versus all other cysteine traps).

Metabolic stability is a function of the whole molecule and cannot elevate the warhead alone as conferring universal property improvement. I agree aldehydes specifically are easily beaten on the chemical stability front.

P10 L298 The RAY1216 6 hour concentrations in mouse are indeed an improvement over PF-7321332. The mouse work is only for efficacy and is not a predictor of human efficacy of single agents (which is the implication, but not relevant). Paxlovid is a combination, ritonavir-boosted two drug therapy. What are the predicted 12 or 24 hour C_{min}, unbound concentrations likely to be for RAY1216 at a viable human dose, and how do they relate to EC₉₀?

P10 L318 How will the superior drug-target residence time manifest itself in patients versus Paxlovid (as opposed to pre-clinically versus PF-7321332)? Why is there the slight difference in viral load lowering for each molecule at 600mg/kg QD? There is not much discussion in the discussion section.

P11 L325 Boceprevir should be drawn as a scrambled stereochemistry at P1 as it is registered as a mixture of stereoisomers.

In vitro data on metabolism (HLM, Hepatocytes), permeability (Caco-2/RRCK.PAMPA), plasma protein binding, SARS-CoV-2 in other cell lines (eg dNHBE), activity against other human coronaviruses, off targets (eg human cathepsins) for RAY1216 would have enriched this opportunity to share a full pre-clinical profile. This appears to be on the way in a separate manuscript on the molecular design process.

Reviewer #2 (Remarks to the Author):

This is a fine manuscript covering so many aspects of developing this alpha-ketoamide into a drug. Really state of the art.

The English should be improved.

Reviewer #3 (Remarks to the Author):

The aim of this paper was to create a SARS-CoV-2 antiviral compound that is both effective and resistant to developing resistance. SARS-CoV-2 is able to evade antibody immunity through antigenic drift, so small molecule drugs can provide an alternative intervention to control COVID-19 pandemic. The authors discuss the development of an inhibitor that targets the SARS-CoV-2 main protease, using an α -ketoamide based peptidomimetic approach.

The new compound was developed using the successful HCV protease inhibitor discovery program used for telaprevir as inspiration. The compound contains distinct chemical elements that make it target the coronavirus M_{pro} more effectively. It also consists of the α -ketoamide active moiety, which has been an essential feature for the efficiency of HCV protease inhibitors. The authors analyzed various enzymatic parameters, such as inhibition

parameters, kinetics, and affinity. The compound has been found to inhibit recombinant Mpro at low concentration (with a K_i of 8.6 nM), and the dissociation constant is lower than that of other Mpro inhibitors currently in use in clinics. Compared to PF-07321332 (nirmatrelvir/Paxlovid), RAY1216 has a longer drug-target residence time of 104 minutes instead of just 9 minutes. The measurements indicate that RAY1216 dissociates around 12 times more slowly than PF-07321332. The authors examined the cause of this slow dissociation by analyzing the crystal structure of the SARS-CoV-2 Mpro:RAY1216 complex. They report that RAY1216 is covalently linked to the catalytic Cys145 through the α -ketoamide active group. Furthermore, the crystal structure indicates that there are more significant interactions between the bound RAY1216 and the Mpro active site as compared to PF-07321332. However, the drug's interaction with the enzyme may differ in infected cells compared to the tight association observed in the crystal structure, as crystal structures are unable to report on the dynamics of the enzyme in solution. It would be interesting to discuss the value of the structural data in the context of the enzymological information.

From can determine, the enzymology and crystallography have been completed adequately, and they provide valuable insights into the drug's mode of action. I will refrain from making any extensive comments on these studies, as individuals with more expertise could offer more beneficial feedback to enhance the report.

In cell culture RAY1216 demonstrates comparable antiviral activities towards different SARS-CoV-2 virus variants compared to PF-07321332. To improve clarity, I suggest enhancing the documentation of the antiviral activity. Currently, only the MTT cell viability assay is being used. However, to better gauge the compound's effectiveness, it would be useful to also determine virus titers and the accumulation of viral RNA through RT-PCR. Additionally, conducting tests on various cell lines could help establish that the antiviral activity is not specific to a particular type of cell. To ensure that the drug being developed will not be rendered ineffective due to the development of resistance, it will be necessary to investigate if the SARS-CoV2 replicons, used for safety reasons, can develop drug resistance. The animal testing of RAY1216 has been well-documented, but it will be important to compare it's in vivo activity to that of PF-07321332. RAY1216 has shown better pharmacokinetics than PF-07321332 in multiple animal models. This could potentially make it a more efficient option.

Author Rebuttal to Initial comments

***** Reviewer Expertise:

Referee #1: Pharmacology Referee #2: Mpro,
Structure Referee #3: Virology

Reviewer Comments:

Reviewer #1 (Remarks to the Author):

The manuscript from Chen, Huang and Kuzmic describes the pre-clinical characterization of RAY1216, a clinical candidate currently in phase 3 clinical trials for the treatment of COVID-19. Given that this is the first disclosure of such an advanced clinical molecule, this represents a potential publication of significance. The paper is extremely well written and referenced, drawing from, and reflecting a field that has expanded enormously over the last 18 months. The enzymology section is of particular note and excellence. The slow on/off characteristics for RAY1216 represents an interesting orthogonality that actually merited greater discussion of potential advantages or disadvantages to the eventual clinical profile of the molecule than was offered.

Throughout the manuscript characterizes RAY1216 alongside PF-07321332, the prototype Mpro inhibitor and the anti-viral component of Paxlovid, the first authorized oral Mpro inhibitor for COVID-19.

I would ask the editor to consider the following in correspondence with the authors before their final decision on whether to publish. While the paper contains all the components one would expect of a manuscript, a number of clarifications/responses are needed before I would support publication.

Proof reading/clarifications:

- 1 P2 L96 either 'in a phase 3 clinical trial' or 'in phase 3 clinical trials'

We thank the reviewer for the suggested wordings, RAY1216 has been approved by the National Medical Products Administration of China in April with the

commercial name “Leritrevir” for COVID-19 treatment, and we have updated its status in the *Abstract, Introduction* and *Conclusion* sections.

2 P6 177 This is a beautiful section on the enzymology. Having differentiated RAY1216 from PF-7321332 at the on/off rate level, how might this help or hinder the in vivo or clinical performance of RAY1216 versus Paxlovid? The manuscript would benefit from such a discussion.

We thank the reviewer for this comment. In our opinion, it would be largely speculative at this stage to suggest a direct link between on/off rate constants and in vivo clinical performance of the two drugs investigated here. This is especially true in light of the antiviral experimental data, such as those in Table

2. Indeed, the Vero E6 cell antiviral titres listed in **Table 2** show that PF- 07321332 has a slight advantage over RAY1216 in experiments involving seven different virus strains. However, we do agree with the reviewer that the manuscript would benefit from directly addressing this appealing but also exceedingly complex topic (Copeland, 2006, doi:10.1038/nrd2082). To this end, we had modified the closing paragraph of the main text by adding the following statement: "*These results suggest that the drug-target residence time alone, as determined in biochemical kinetic assays, can not solely dictate pharmacological efficacy.*".

3. P8 L244 The anti-viral work is conducted in VeroE6 cells. In the Science paper by Owen et al on the characterization of PF-7321332, it is stated that this assay must be conducted in the presence of an efflux inhibitor (EI) CP-100356. There is no mention of it in the experimental procedure or figure legends. PF-7321332 is a significant Pgp substrate and anti-viral activity cannot be seen with the presence of the EI when using the VeroE6 system. 1. Did the authors use the EI? 2. If not, how is the control PF-7321332 data explained? 3. Is RAY1216 a Pgp substrate (would seem likely)?

We thank the reviewer for this question, we did use the CP-100356 efflux inhibitor for the anti-viral experiments using Vero E6 cells. However, unfortunately, by accident, we failed to include this information in our original method section. To rectify this omission, we have added sentences in the method sections - "*Cytotoxicity and cytopathic effect (CPE) inhibition assay*", "*Plaque-reduction assay*" and "*Virus inhibition assay by qPCR*" to indicate that 2 μ M efflux inhibitor (EI) CP-100356 was used when Vero E6 cells were used for antiviral study.4. P9 L275 Is plasmata a word? I looked up if was some type of plural of plasma. Did not appear to be, but I may stand corrected!

We have changed the word to “plasmas”.

5. P8/9 In vivo model. It looks the authors decided to compare the two molecules at the same 600mg/kg QD dose. With different PK, potentially PD (due to on/off rate differences) and to a lesser degree potency, this is not the ideal way to compare efficacy for the two molecules. These were not the doses/dosing regime used for PF- 7321332 in the comparative paper cited. As a result, RAY1216 looks less effective in viral load lowering, albeit not to significance in this study from Chen. PF- 7321332 is capable of offering up to a 2.5 log unit drop in this model when dosed 1000mg/kg BID.

We thank the reviewer for this comment, due to local P3 laboratory regulations, we are only allowed to enter the lab once daily with a ~ 4-hour time slot. In addition, due to formulation limitation, each dose of RAY1216 is limited at 600 mg/mg. Due to the above limitations, we were only able to test this dosage and perform parallel experiments with PF-7321332 under our current set-up.

6. What is the maximum efficacy for RAY1216 at doses above 600mg/kg? Did toxicity/formulation preclude higher dosing?

Due to our formulation limitation (solubility), the maximum dose for each oral administration was limited at 600 mg/kg for animal experiments.

6. continued. The authors should provide a mouse free drug exposure plot for both molecules over 24 hours at 600mg/kg QD with a reference concentration line of the VeroE6 EC90 for both molecules. How much time does each molecule spend above EC90 as free drug and if so, at what are the multiples over EC90 at 12 and 24hours?

We thank the reviewer for this suggestion, we have performed the suggested pharmacokinetics experiments with the K18 h-ACE2 transgenic mouse using 600 mg/kg QD. The results are shown in Fig. S17 and Table S10. The data show that under the same condition, PF-7321332 can maintain plasma concentration above EC90 (EC90 of Delta variant by CPE method was usedbecause Delta variant was used in the animal antiviral study) for ~ 5 hours, while RAY1216 can maintain plasma concentration above EC90 for at least 8 hours (from the extrapolated curve, RAY1216 should maintain plasma concentration above EC90 for ~ 20 hours). The AUC for RAY1216 is approximately 6 times larger than that of PF-7321332, these data are largely consistent with PK data obtained at lower administered doses for both drugs.

6. continued. This assumes that 600mg/kg PK is available for each molecule (Ideally satellite non-infected animals given the same dosing solution as the infected ones). If not, it should really be run. The 10mg oral mouse PK stated in the paper, rightly referencing multiples of EC90, is useful but the PK component of the doses used in the efficacy are critical to interpret the in vivo outcome.

Following the reviewer's suggestion, we have done the suggested PK experiment with K18 h-ACE2 transgenic mouse using an oral dose of 600mg/kg QD and the results are shown in Fig. S17 and Table S10.

7. Do the slow/on off characteristics have any connection to the in vivo efficacy seen?

We thank the reviewer for this question and we have attempted to answer this question in Q2. We should add that, although RAY1216 has slower off-rate and slower metabolism than PF-7321332, it is puzzling that we only observed comparable but slightly less favourable antiviral effects for RAY1216 in cell culture and mouse model, we stressed that in the discussion section "On the other hand, PF-07321332 is slightly favoured over RAY1216 in reducing mouse lung viral titre." further investigation is likely required for better understanding of such observations.

8. In the bioanalysis did the authors see any epimerization of the molecule at the P1 center? Boceprevir is a mixture of diastereomers at P1 given the stereochemical instability of an α -ketoamide in that case. Was RAY1216 stable to epimerization at P1? It seems to have been synthesized as a single stereoisomer in the experimental section, but what is the stability in blood or overtime? This will affect the interpretation of the in vivo study as it could be a non- metabolic source of active compound decline. It is surprising that there is no mention of this potential property for α -ketoamides in the manuscript.

We thank the reviewer for this suggestion, we synthesised the P1 R-epimer of RAY1216 called "RAY1216-E" and we used it as standard in LC-MS/MS analysis to allow investigation of epimerization of RAY1216 at its P1 stereocenter in plasmas of various species. The detailed method is updated in the method section "*Stability of RAY1216 and epimerization of RAY1216*". The results (updated in Fig. S16 and Table S9), show that, in mouse, rat and human plasmas, the P1 S-epimer of RAY1216 is relatively stable with half-lives longer than 4 hours. In dog and monkey plasmas, RAY1216 is less stable showing half-lives of 82 min and 147 min. After 2 hr incubation in plasma, 4% (mouse), 2.6% (rat), 51.9% (dog), 13.2% (monkey), 10.6% (human) of RAY1216 was converted to P1 R-epimer, respectively. Our preliminary data show that the P1 R-epimer can also inhibit M^{pro} enzyme activity but with an IC50 4 times weaker.

9. P9 L287 This was the only sentence that was a little sweeping or unsupported in the entire manuscript. It is difficult to credit a-ketoamides as having superior chemical and metabolic stability (implying versus all other cysteine traps). Metabolic stability is a function of the whole molecule and cannot elevate the warhead alone as conferring universal property improvement. I agree aldehydes specifically are easily beaten on the chemical stability front.

We thank the reviewer for this comment and we withdraw this sentence.

10. P10 L298 The RAY1216 6 hour concentrations in mouse are indeed an improvement over PF-7321332. The mouse work is only for efficacy and is not a predictor of human efficacy of single agents (which is the implication, but not relevant). Paxlovid is a combination, ritonavir- boosted two drug therapy. What are the predicted 12 or 24 hour C_{min}, unbound concentrations likely to be for RAY1216 at a viable human dose, and how do they relate to EC90?

In the clinical trial, several dosings have been tested without ritonavir. A single administration of RAY1216 800 mg p.o. with high-fat diet can attain a C_{max} of 8510 ng/ml with a T_{1/2} of 7 hr. (versus C_{max}=4760 ng/ml, T_{1/2} of 7 hr, attained under fasting condition), these are well above the PPB corrected EC₉₀ = 619 ng/ml [EC₉₀ determined by CPE for Omicron BA.1 (208 nM = 133 ng/mL) corrected for plasma protein binding (PPB) = 133/0.287(human PPB) = 463 ng/mL]. In the recommended clinical dosing condition (RAY1216 400mg TID

15is administered for 5 days), C_{max} can reach 3450 ng/ml with a $T_{1/2}$ of 2.3 hr; the average steady-state blood concentrations at 6 h, 12 h and 24 h after the first administration are 965 ng/ml, 1205 ng/ml and 425 ng/ml, respectively. These concentrations are ~ 2.1 -fold, ~ 2.6 -fold and ~ 0.9 fold of the PPB corrected EC_{90} for Omicron BA.1. Some of the above quoted clinical dosing data are given in the “Letrivevir” product manual. A manuscript is also being prepared to disclose relevant clinical PK data in an academic journal.

11. P10 L318 How will the superior drug-target residence time manifest itself in patients versus Paxlovid (as opposed to pre-clinically versus PF-7321332)? Why is there the slight difference in viral load lowering for each molecule at 600mg/kg QD? There is not much discussion in the discussion section.

We thank the reviewer for this question. A correlation between drug-residence time and pharmacological effect has been discussed (Copeland, 2006, doi:10.1038/nrd2082) and we agree that *“it has recently emerged that drug-target residence time is an important parameter to optimise for drug efficacy (Copeland et al., 2006; Dahl and Akerud, 2013; Lu and Tonge, 2010)”* in our discussion. At this moment, in the case of RAY1216, we hesitate to extrapolate drug-target residence time to clinical benefit, although it is a very interesting feature of the drug. In the discussion, we added the following comments: *“RAY1216 possesses superior drug-target residence time and pharmacokinetic properties when compared with PF-07321332 (nirmatrelvir), the active anti-viral component in Paxlovid. On the other hand, PF-07321332 is slightly favoured over RAY1216 in reducing mouse lung viral titre. These results suggest that the drug-target residence time alone, as determined in biochemical kinetic assays, can not solely dictate pharmacological efficacy.”* It should also be noted that being a single-component drug without ritonavir, RAY1216 can potentially avoid unwanted drug-drug interactions known to be caused by ritonavir in clinical use.

12. P11 L325 Boceprevir should be drawn as a scrambled stereochemistry at P1 as it is registered as a mixture of stereoisomers.

We thank the reviewer for this comment, we have redrawn the chemical structure of Boceprevir in Fig.1 with a wavy line at P1.

13. In vitro data on metabolism (HLM, Hepatocytes), permeability (Caco-

2/RRCK.PAMPA), plasma protein binding, SARS-CoV-2 in other cell lines (eg dNHBE), activityagainst other human coronaviruses, off targets (eg human cathepsins) for RAY1216 would have enriched this opportunity to share a full pre-clinical profile. This appears to be on the way in a separate manuscript on the molecular design process.

We thank the reviewer for the suggestions, we are planning to include most of the said data in a separate report focusing on the molecular design process. We do follow the suggestions to include the specificity data of RAY1216 against other human proteases in Table S2. RAY1216 inhibits hCathepsin B, hChymotrypsin C, hCathepsin D and hElastase with IC50s of 4.6 μ M, >100 μ M, >100 μ M and 85 μ M, respectively. Under the same condition, RAY1216 inhibits SARS-CoV-2 M^{PRO} with an IC50 of 9 nM.

Reviewer #2 (Remarks to the Author):

This is a fine manuscript covering so many aspects of developing thus alpha-ketoamide into a drug. Really state of the art.

The English should be improved.

We thank the reviewer for the favourable comments. We have revised various sentences throughout the text and we will continue to improve our English writing during the revision process.

Reviewer #3 (Remarks to the Author):

The aim of this paper was to create a SARS-CoV-2 antiviral compound that is both effective and resistant to developing resistance. SARS-CoV-2 is able to evade antibody immunity through antigenic drift, so small molecule drugs can provide an alternative intervention to control COVID-19 pandemic. The authors discuss the development of an inhibitor that targets the SARS-CoV-2 main protease, using an α -ketoamide based peptidomimetic approach.

The new compound was developed using the successful HCV protease inhibitor discovery program used for telaprevir as inspiration. The compound contains distinct chemical elements that make it target the coronavirus M^{pro} more effectively. It also consists of the

α -ketoamide activemoiety, which has been an essential feature for the efficiency of HCV protease inhibitors. The authors analyzed various enzymatic parameters, such as inhibition parameters, kinetics, and affinity. The compound has been found to inhibit recombinant Mpro at low concentration (with a K_i of 8.6 nM), and the dissociation constant is lower than that of other Mpro inhibitors currently in use in clinics. Compared to PF-07321332 (nirmatrelvir/Paxlovid), RAY1216 has a longer drug-target residence time of 104 minutes instead of just 9 minutes. The measurements indicate that RAY1216 dissociates around 12 times more slowly than PF-07321332. The authors examined the cause of this slow dissociation by analyzing the crystal structure of the SARS-CoV-2 Mpro:RAY1216 complex. They report that RAY1216 is covalently linked to the catalytic Cys145 through the α -ketoamide active group. Furthermore, the crystal structure indicates that there are more significant interactions between the bound RAY1216 and the Mpro active site as compared to PF-07321332. However, the drug's interaction with the enzyme may differ in infected cells compared to the tight association observed in the crystal structure, as crystal structures are unable to report on the dynamics of the enzyme in solution. It would be interesting to discuss the value of the structural data in the context of the enzymological information.

From can determine, the enzymology and crystallography have been completed adequately, and they provide valuable insights into the drug's mode of action. I will refrain from making any extensive comments on these studies, as individuals with more expertise could offer more beneficial feedback to enhance the report.

1. In cell culture RAY1216 demonstrates comparable antiviral activities towards different SARS-CoV-2 virus variants compared to PF-07321332. To improve clarity, I suggest enhancing the documentation of the antiviral activity. Currently, only the MTT cell viability assay is being used. However, to better gauge the compound's effectiveness, it would be useful to also determine virus titers and the accumulation of viral RNA through RT-PCR.

We thank the reviewer for this suggestion, we performed additional plaque reduction assays and qPCR assays to assess virus inhibition. The results are now updated in Fig.4a, Fig.S14, Fig.S15, Table.2 and Table.S8. Consistent with previous conclusion, new data show that RAY1216 and PF-07321332 have comparable antiviral activities towards various strains of SARS-CoV-2.

2. Additionally, conducting tests on various cell lines could help establish that the antiviral activity is not specific to a particular type of cell. To ensure that the drug being developed will not be rendered ineffective due to the development of resistance, it will be necessary to investigate if the SARS-CoV2 replicons, used for safety reasons, can develop drug resistance.

We thank the reviewer for this suggestion, our current local regulations strictly forbid generation of artificial mutations in SARS-CoV-2 live virus, following the reviewer's suggestion we used a cell line carrying a previously reported replicon system (<https://doi.org/10.1007/s12250-021-00385-9>) to investigate whether RAY1216 can induce M^{pro} mutations. In this replicon system, the region encoding S protein is replaced with a luciferase gene in the SARS-CoV-2 genome. The modified genome is placed under the control of a CMV promoter for RNA synthesis in cell. We linearised the vector carrying this replicon before transfecting it into HEK293T cells. After verifying that luciferase was expressed, we passaged the cells in the presence of 50 nM RAY1216, which is the EC50 concentration determined for the replicon in HEK293T cell. The cells carrying the replicon were passaged for 25 generations in a duration of 9 weeks. At the end of the passage, luciferase activity could still be measured. RNA from the cells were reverse transcribed and subjected to next-generation sequencing in the nsp5 region. Unfortunately, the sequencing did not identify any meaningful mutations in nsp5. Although we were not able to use RAY1216 to induce mutations, we used a different replicon system (doi: [10.1128/mBio.02754-20](https://doi.org/10.1128/mBio.02754-20)), allowing easier manipulation of M^{pro}, to study several reported mutations induced by PF-07321332 and their effects on RAY1216 inhibition. RAY1216 is somewhat different in sensitivity towards these mutations. We updated the results in the new section "*M^{pro} mutants and RAY1216 inhibition*" with Fig.6 and Table.4.

The animal testing of RAY1216 has been well-documented, but it will be important to compare it's in vivo activity to that of PF-07321332. RAY1216 has shown better pharmacokinetics than PF-07321332 in multiple animal models. This could potentially make it a more efficient option.

We thank the reviewer for the comment. We have expanded our discussion: *“In summary, RAY1216 possesses superior drug-target residence time and pharmacokinetic properties when compared with PF-07321332 (nirmatrelvir), the active anti-viral component in Paxlovid. On the other hand, PF-07321332 is slightly favoured over RAY1216 in reducing mouse lung viral titre. These results suggest that the drug-target residence time alone, as determined in biochemical kinetic assays, can not solely dictate pharmacological efficacy”*. It should also be noted that being a single-component drug without ritonavir, RAY1216 can potentially avoid unwanted drug-drug interactions known to be caused by ritonavir in clinical use. In this respect, we stressed that “Leritrelvir” is a single-component drug in the text. It was originally found that in animal PK studies to be more stable than PF-07321332 and such observation inspired its trial as a single-component drug. We mentioned this point in our discussion *“In pharmacokinetic studies, RAY1216 showed improved elimination half-lives compared to PF-07321332. This may allow its use without ritonavir which is known to have significant unwanted drug-drug interactions.”*.

Other changes:

1. We updated the mouse and rat PK data with 5 male and 5 female animals for each drug to give better statistics confidence. (updated in **Fig. 5**)
2. We confirm that RAY1216 forms a more stable inhibition complex of M^{Pro} in solution by differential scanning fluorimetry (DSF). Binding of RAY1216 increases melting temperature of M^{Pro} by 20 °C, while binding of PF-07321332 increases M^{Pro} melting temperature by 11 °C. (updated in **Fig. S10** and **Table S6**)
3. We used enzyme inhibition kinetics to study M^{Pro} P132H mutant which is dominant in current circulating strains. The results show that both RAY1216 and PF-07321332 maintain similar levels of inhibition compared to WT M^{Pro} (updated in **Fig. S18** and **Table S11**).

Decision Letter, first revision:

Message: Our ref: NMICROBIOL-23020441A

18th December 2023

Dear Xiaoli,

Thanks for your email and for your patience as we've prepared the guidelines for final submission of your Nature Microbiology manuscript, "Inhibition mechanism and antiviral activity of RAY1216, an α -ketoamide based SARS-CoV-2 main protease inhibitor" (NMICROBIOL-23020441A). Please carefully follow the step-by-step instructions provided in the attached file, and add a response in each row of the table to indicate the changes that you have made. Please also check and comment on any additional marked-up edits we have proposed within the text. Ensuring that each point is addressed will help to ensure that your revised manuscript can be swiftly handed over to our production team.

In recognition of the time and expertise our reviewers provide to Nature Microbiology's editorial process, we would like to formally acknowledge their contribution to the external peer review of your manuscript entitled "Inhibition mechanism and antiviral activity of RAY1216, an α -ketoamide based SARS-CoV-2 main protease inhibitor". For those reviewers who give their assent, we will be publishing their names alongside the published article.

Nature Microbiology offers a Transparent Peer Review option for new original research manuscripts submitted after December 1st, 2019. As part of this initiative, we encourage our authors to support increased transparency into the peer review process by agreeing to have the reviewer comments, author rebuttal letters, and editorial decision letters published as a Supplementary item. When you submit your final files please clearly state in your cover letter whether or not you would like to participate in this initiative. Please note that failure to state your preference will result in delays in accepting your manuscript for publication.

2Cover suggestions

COVER ARTWORK: We welcome submissions of artwork for consideration for our cover. For more information, please see our [guide for cover artwork](https://www.nature.com/documents/Nature_covers_author_guide.pdf).

Nature Microbiology has now transitioned to a unified Rights Collection system which will allow our Author Services team to quickly and easily collect the rights and permissions required to publish your work. Approximately 10 days after your paper is formally accepted, you will receive an email in providing you with a link to complete the grant of rights. If your paper is eligible for Open Access, our Author Services team will also be in touch regarding any additional information that may be required to arrange payment for your article.

Please note that *Nature Microbiology* is a Transformative Journal (TJ). Authors may publish their research with us through the traditional subscription access route or make their paper immediately open access through payment of an article-processing charge (APC). Authors will not be required to make a final decision about access to their article until it has been accepted. [Find out more about Transformative Journals](https://www.springernature.com/gp/open-research/transformative-journals)

Authors may need to take specific actions to achieve [compliance with funder and institutional open access mandates](https://www.springernature.com/gp/open-research/funding/policy-compliance-faqs). If your research is supported by a funder that requires immediate open access (e.g. according to [Plan S principles](https://www.springernature.com/gp/open-research/plan-s-compliance)) then you should select the gold OA route, and we will direct you to the compliant route where possible. For authors selecting the subscription publication route, the journal's standard licensing terms will need to be accepted, including [self-archiving policies](https://www.nature.com/nature-portfolio/editorial-policies/self-archiving-and-license-to-publish). Those licensing terms will supersede any other terms that the author or any third party may assert apply to any version of the manuscript.

Until then, I wish you and your co-authors happy Christmas and a happy new year!

Best wishes,

Reviewer #1:

Remarks to the Author:

Thank you for your polite responses to my requests as a reviewer.

Reviewer #3:

Remarks to the Author:

The authors have diligently addressed all reviewers' comments by making numerous additions to the manuscript. Notably, their inclusion of experiments investigating the potential development of resistance to RAY1216 is commendable. Additionally, the authors have provided several clarifications to highlight the comparative effectiveness of this novel drug in preclinical models when compared to clinically approved SARS-CoV2 protease inhibitors. In my opinion, this study is robust and holds promise as an alternative antiviral strategy.

Final Decision Letter:

Mes 22nd January 2024

sag

e: Dear Xiaoli,

I am pleased to accept your Article "Preclinical evaluation of the SARS-CoV-2 Mpro inhibitor RAY1216 with improved pharmacokinetics compared to nirmatrelvir" for publication in Nature Microbiology. Thank you for having chosen to submit your work to us and many congratulations.

You may wish to make your media relations office aware of your accepted publication, in case they consider it appropriate to organize some internal or external publicity. Once your paper has

4been scheduled you will receive an email confirming the publication details. This is normally 3-4 working days in advance of publication. If you need additional notice of the date and time of publication, please let the production team know when you receive the proof of your article to ensure there is sufficient time to coordinate. Further information on our embargo policies can be found here: <https://www.nature.com/authors/policies/embargo.html>

Please note that *Nature Microbiology* is a Transformative Journal (TJ). Authors may publish their research with us through the traditional subscription access route or make their paper immediately open access through payment of an article-processing charge (APC). Authors will not be required to make a final decision about access to their article until it has been accepted. [Find out more about Transformative Journals](https://www.springernature.com/gp/open-research/transformative-journals)

Authors may need to take specific actions to achieve [compliance](https://www.springernature.com/gp/open-research/funding/policy-compliance-faqs) with funder and institutional open access mandates. If your research is supported by a funder that requires immediate open access (e.g. according to [Plan S principles](https://www.springernature.com/gp/open-research/plan-s-compliance)) then you should select the gold OA route, and we will direct you to the compliant route where possible. For authors selecting the subscription publication route, the journal's standard licensing terms will need to be accepted, including [self-archiving policies](https://www.nature.com/nature-portfolio/editorial-policies/self-archiving-and-license-to-publish). Those licensing terms will supersede any other terms that the author or any third party may assert apply to any version of the manuscript.

An online order form for reprints of your paper is available at <https://www.nature.com/reprints/author-reprints.html>. All co-authors, authors' institutions and authors' funding agencies can order reprints using the form appropriate

to their geographical region.

Congrats to you and your co-authors again! I'm looking forward to seeing your paper published.

Best wishes,

P.S. Click on the following link if you would like to recommend Nature Microbiology to your librarian <http://www.nature.com/subscriptions/recommend.html#forms>

** Visit the Springer Nature Editorial and Publishing website at http://editorial-jobs.springernature.com?utm_source=ejP_NMicro_email&utm_medium=ejP_NMicro_email&utm_campaign=ejP_NMicro for more information about our career opportunities. If you have any questions please click [here](mailto:editorial.publishing.jobs@springernature.com).**